# Wave-induced stress and breaking of sea ice in a coupled hydrodynamic–discrete-element wave–ice model

Agnieszka Herman[1]

[1]Institute of Oceanography, University of Gdansk, Poland

*Correspondence to:* A. Herman (oceagah@ug.edu.pl)

**Abstract.** In this paper, a coupled sea ice–wave model is developed and used to analyze wave-induced stress and breaking in sea ice for a range of wave and ice conditions. The sea ice module is a discrete-element bonded-particle model, in which ice is represented as cuboid "grains" floating on the water surface that can be connected to their neighbors by elastic joints. The joints may break if instantaneous stresses acting on them exceed their strength. The wave module is based on an open-source

version of the Non-Hydrostatic WAVE model (NHWAVE). The two modules are coupled with proper boundary conditions for pressure and velocity, exchanged at every wave model time step. In the present version, the model operates in two dimensions (one vertical and one horizontal) and is suitable for simulating compact ice in which heave and pitch motion dominates over surge. In a series of simulations with varying sea ice properties and incoming wavelength it is shown that wave-induced stress reaches maximum values at a certain distance from the ice edge. The value of maximum stress depends on both ice properties

and characteristics of incoming waves, but, crucially for ice breaking, the location at which the maximum occurs does not change with the incoming wavelength. Consequently, both regular and random (Jonswap spectrum) waves break the ice into floes with almost identical sizes. The width of the zone of broken ice depends on ice strength and wave attenuation rates in the ice.

## 1 Introduction

Interactions between sea ice and waves are a defining characteristic of the marginal ice zone (MIZ), loosely defined as a region of the sea ice cover adjacent to the ice edge and directly influenced by the neighboring open ocean. In recent years, as the sea ice extent in polar and subpolar regions of the northern hemisphere decreases and thick, multi-year ice is replaced with thinner, weaker seasonal ice, conditions typical for MIZ (ice concentration lower than 90%, small floe sizes, patchy distribution of floes on the sea surface, etc.) tend to occur over larger and larger areas. There is a growing observational and modeling evidence

that wave–ice interactions play an important role in the observed expansion of MIZ and negative trends in sea ice extent (see, e.g., Asplin et al., 2012, 2014; Thomson and Rogers, 2014; Thomson et al., 2016). Thin, fragmented sea ice is susceptible to further breaking and, depending on ambient weather and oceanic conditions, melting, which facilitates faster ice drift, decrease in ice concentration, increase in wind fetch, and thus creates more favorable conditions for wave propagation and generation, leading to still stronger fragmentation. These – and many other – feedbacks suggest that it is crucial to include (the effects of)

wave–ice interactions in numerical ocean–sea ice–atmosphere models in order to be able to reliably reproduce the observed

processes and forecast future changes on both synoptic and climate scales. Parameterizations of wave–ice interactions for large-scale, continuum models (i.e., those in which ice is treated as a continuous mass rather than as discrete particles) are crucial for further development of those models. However, although appreciable effort has been made in that direction in recent years (Dumont et al., 2011; Doble and Bidlot, 2013; Squire et al., 2013; Williams et al., 2013, 2017; The WAVEWATCH III®

Development Group (WW3DG), 2016; Bennetts et al., 2017), our understanding of many aspects of wave–ice interactions is still too limited to allow formulating such parameterizations, especially those suitable for a wide range of conditions. Strong fragmentation of the ice into many small floes, and highly energetic environment due to the presence of waves make the MIZ a very difficult, demanding location for field work. Due to their low temporal resolution in polar regions, satellite data only provide snapshots of sea ice conditions, making it difficult or impossible to infer details of processes acting on time scales

comparable with a typical wave period. Therefore, in spite of recent advances in remote sensing techniques to monitor waves in the MIZ (e.g., Ardhuin et al., 2017), the amount of observational data necessary for validation of numerical models remains very limited. Consequently, many crucial processes and their large-scale effects are only poorly understood. As the overview of the relevant literature in the following paragraphs clearly shows, one of them is sea ice breaking by waves and the resulting floe-size distribution (FSD) – the main subject of this paper.

Review papers by Squire et al. (1995) and Squire (2007, 2011) provide a good overview of the state-of-the-art research related to wave–ice interactions. Problems studied in this context include, but are not limited to: wave propagation, attenuation and scattering by various ice types, e.g., continuous ice sheets, broken compact ice, (groups of) individual ice floes, and inhomogeneities like pressure ridges, cracks etc.; motion of ice floes (and other floating objects, including very large floating structures) on waves and wave-induced floe collisions; sea ice breaking by waves. Considering relatively rich literature on wave

propagation in sea ice and wave-induced motion of ice floes/sheets (see, e.g., Squire, 1983; Liu and Mollo-Christensen, 1988; Shen and Ackley, 1991; Meylan and Squire, 1994; Meylan, 2002; Wang and Shen, 2011; Montiel et al., 2012, 2016; Sutherland and Rabault, 2016, and references there, as this list is by far not complete), the number of studies on sea ice breaking by waves is remarkably limited and – as Squire et al. (1995) aptly put it – they are to a large degree based on "anecdotal evidence". In a series of papers published in 1980s, V. Squire analyzed wave propagation in continuous, land-fast ice and basic mechanisms

of wave-induced ice breaking, related to the presence of secondary ice-coupled waves affecting the wave envelope close to the ice edge and rapidly decaying away from the edge (see, e.g., Squire, 1984a, b). In their review paper, Squire et al. (1995) describe qualitatively the process of breaking of land-fast ice by swell waves, in which elongated, parallel strips of ice are progressively separated from the initially continuous ice sheet. They write that "the width of the strips, and hence the diameter of the floes created by the process, is remarkably consistent and appears in the sparse evidence available to be rather insensitive

to the spectral structure of the sea, but highly dependent on ice thickness." Consistently, their modeling results showed that the location of maximum flexural strain in the ice relative to the ice edge depends mainly on ice thickness rather than wave period. Notwithstanding these conclusions, a close relationship between the incoming wavelength and floe sizes produced by breaking is usually assumed, as for example in the above-mentioned parameterizations by Williams et al. (2013), Bennetts et al. (2017) and others. It is worth stressing that these models do not directly simulate the sea ice breaking process. Instead,

they simulate the *effects* of breaking by testing if the conditions are favorable for breaking (criteria based on the wave height

and thus strain that the ice experiences) and, if these conditions are fulfilled, by modifying the maximum floe size $L_{\max}$ according to certain, prescribed rules. The shape of the FSD for floe sizes $L_o < L_{\max}$ is prescribed as well, so that $L_{\max}$ is the only variable parameter characterizing the FSD. In other words, these models are suitable for analyzing the consequences of wave-induced breaking of sea ice (i.e., the influence of the evolving FSD on ice dynamics and/or thermodynamics), given the assumed relationships between the FSD and the wave forcing. Thus, as with any parameterization, our understanding of the processes involved decides upon the validity and accuracy of the modelling results.

Since the pioneering works described above, few studies have been devoted specifically to the analysis of sea ice breaking by waves. In a modeling study of ice motion on waves, Meylan and Squire (1994) analyzed flexural strain variability in ice floes of different sizes and thicknesses. Langhorne et al. (1998) analyzed experimentally and numerically the fatigue behavior of first-year sea ice subject to repeated bending stress and demonstrated that the time history of strain acting on the ice is crucial for predicting its breaking. In a subsequent work, Langhorne et al. (2001) extended their earlier work to estimate lifetime of landfast ice subject to waves with given characteristics. Based on ship observations of ice breaking during a strong-wave event in the Barents Sea, Collins et al. (2015) analyzed the role of nonlinear wave processes and the resulting strong modulation of wave amplitude in ice breaking, in accordance with much earlier observations and theoretical results of Liu and Mollo-Christensen (1988). Vaughan and Squire (2011) estimated ice breaking probabilities in the Arctic sea ice in function of the distance from the ice edge, based on the probability density functions of the sea surface curvature. This approach, employed also by Kohout and Meylan (2008), assumes a simple relationship between strain (estimated directly from the shape of the wave profile) and stress in the ice. Finally, sea ice breaking is included in the recent model of wave–sea ice interactions by Montiel and Squire (2017). In simulations of wave propagation and multiple scattering by circular ice floes in MIZ, they used strain-based floe breaking criteria and obtained approximately normal FSDs, without any *a priori* assumptions regarding their shape.

In this paper, a coupled sea ice–wave model is proposed suitable for simulating ice–wave interactions in the time domain, including computation of instantaneous stresses in ice and ice breaking. The model consists of a bonded-particle discrete-element sea ice model, similar to that of Herman (2016), and a wave model based on the code of the Non-Hydrostatic WAVE (NHWAVE) model by Ma et al. (2012, 2014). The two parts are coupled with proper boundary conditions exchanged at every NHWAVE time step. The type of a discrete-element model (DEM) used here, in which bonds connecting grains behave as elastic "rods", is particularly suitable for studying sea ice–wave interactions due to oscillatory nature of these processes, prohibiting inelastic effects from becoming significant (see, e.g., Fox and Squire, 1994).

Apart from providing a detailed description of the model, the main goal of this work is, first, to analyze spatiotemporal variability of wave-induced stress in ice floes with varying thickness and sizes, and second, to analyze the time evolution of breaking and the final breaking patterns produced by regular and irregular waves. The paper is structured as follows: Section 2 contains the definitions and assumptions underlying the model, followed by the description of the model equations and coupling between the wave and ice modules. The results of simulations are presented in Section 3. Finally, Section 4 provides a discussion and a summary.

## 2  Model description

The model consists of two parts, the sea ice module and the wave module, exchanging information at every time step. The wave part is based on the Version 2.0 of the Non-Hydrostatic WAVE (NHWAVE) model developed by Ma et al. (2012) and available at `https://sites.google.com/site/gangfma/nhwave`. NHWAVE solves three-dimensional incompressible Navier-Stokes equations in vertically-scaled $\sigma$-coordinates (see further Section 2.2.1). For the purpose of this work, NHWAVE has been extended to allow non-free surface boundary conditions under the (floating) ice, as described in detail further in Section 2.2.3. The second component is a discrete-element bonded-particle sea ice model. It is based on similar ideas and assumptions as the DESIgn model by Herman (2016), with certain modifications crucial for representing ice motion and bending on the oscillating sea surface (in DESIgn, which is essentially two-dimensional in the horizontal plane, these effects are treated in a very rudimentary way, with a number of unrealistic assumptions).

Recently, Ma et al. (2016) and Orzech et al. (2016) implemented in NHWAVE equations for floating objects and other solid "obstacles". Their method is based on immersed boundary techniques (Mittal and Iaccarino, 2005; Ha et al., 2014), suitable for modeling interactions between fully or partially submerged solid bodies (fixed or moving) and the surrounding fluid. The algorithms of Orzech et al. (2016) are not yet included in the publicly available version of NHWAVE (although the code does contain basic treatment of fixed obstacles); the present model, developed independently, shares many features with their approach, but due to a number of assumptions related to the shape and the characteristics of motion of the floating objects, it is much less general, suitable for the specific configuration analyzed in this work. On the other hand, the model of Orzech et al. (2016) assumes that floating objects are rigid bodies, making it unsuitable for an analysis of ice deformation and breaking, crucial for the present study.

### 2.1  Definitions and assumptions

The model is two-dimensional in the $xz$ plane. The waves are unidirectional and propagate along the $x$ axis; the $z$ axis is directed vertically upward, with $z = 0$ at the mean water level. The sea ice is composed of discrete elements (called grains) of cuboidal shape that are floating on the water surface and may be bonded to their neighbors with elastic bonds. The grains are rigid bodies, so that the deformation of the sea ice is accommodated only by the bonds, which may break during the simulation if stresses acting on them exceed their strength.

In the present version of the model it is assumed that the horizontal resolution of the wave model, $\Delta x$, and the sizes of the grains are adjusted, i.e., every one of the $i = 1, \ldots, N_x$ grid cells of the wave model is either ice-free or fully covered with ice (Fig. 1). Let us denote a set of indices of ice-covered cells as $\mathcal{I}_{\mathrm{g}}$. All grain-related variables and equations referenced further are relevant for $i \in \mathcal{I}_{\mathrm{g}}$. Similarly, as bonding is possible only between grains occupying neighboring cells, we may define a set of bond indices $\mathcal{I}_{\mathrm{b}}$ so that $i \in \mathcal{I}_{\mathrm{b}}$ if and only if both $i \in \mathcal{I}_{\mathrm{g}}$ and $(i+1) \in \mathcal{I}_{\mathrm{g}}$. (To avoid renumbering of bonds during a simulation, broken bonds are not removed from the list, but their strength is set to zero – see further.)

The grains have length $2l_i = \Delta x$, thickness $h_i$, and mass density $\rho_i$ (Fig. 2). The model equations are formulated for an ice "strip" with unit width in the $y$ direction. The position of the center of the $i$th grain is $[x_i, z_i]$, and the deviation of its

orientation from the horizontal position due to rotation in the $xz$ plane is denoted with $\theta_i$. The motion of the grains is described by the translational velocity $[u_i, w_i]$ and the angular velocity $\omega_i$. For each grain, the center of mass and the center of rotation are assumed identical, so that the off-diagonal elements of the mass and buoyancy matrices vanish. For rotation within the $xz$ plane, the moment of inertia per unit grain width $I_{g,i} = \rho_i \frac{l_i h_i}{6}(h_i^2 + 4l_i^2)$. The mass per unit grain width is $m_i = 2\rho_i l_i h_i$. The assumption regarding the grains' positions relative to the wave model cells implies that $u_i \equiv 0$ and $x_i$ is constant, which makes the model applicable only to compact sea ice in which the drift and oscillatory surge motion is insignificant (obviously, this is true in a continuous, unbroken ice sheet; in broken ice at high ice concentration, i.e., with densely packed floes, horizontal motion is suppressed by collisions between neighbouring floes). These limitations will be relaxed in the future versions.

All bonds are cuboid (Fig. 2) and their geometric properties are: thickness $h_{b,i}$ and length $l_{b,i} = \lambda(l_i + l_{i+1}) = \lambda \Delta x$, where $\lambda \in (0, 1]$ is a coefficient deciding whether the elastic deformation is distributed across the grains ($\lambda = 1$) or limited to narrow zones at the grains' boundaries ($\lambda \to 0$). As in the case of grains, it is assumed that the bonds have unit widths in the $y$ direction. Additionally, the bonds have the following material properties: Young's modulus $E_{b,i}$, ratio of the normal to shear stiffness $\lambda_{ns,i}$; tensile strength $\sigma_{t,br,i}$; compressive strength $\sigma_{c,br,i}$, and shear strength $\tau_{br,i}$. From this set of properties, the normal and shear stiffness can be calculated: $k_{n,i} = E_{b,i}/l_{b,i}$ and $k_{t,i} = k_{n,i}/\lambda_{ns,i}$, respectively. Finally, the relevant moments of inertia (again, per unit bond width) are $I_{b,i} = \frac{1}{12}h_{b,i}^3$.

Due to the assumption of no motion along the $x$ direction, no contact model is necessary for neighboring grains that are not bonded to each other. (If surge is taken into account, repulsive contact forces between touching grains should be implemented, e.g., the Hertzian model, as used in Herman, 2016).

In the vertical direction, the model domain is bounded by $z = -H(x)$ and $z = \eta(x, t)$, where $H(x)$ denotes the (time-independent) water depth and $\eta(x, t)$ denotes the instantaneous water surface elevation. The total instantaneous water depth is $D(x, t) = H(x) + \eta(x, t)$.

## 2.2 Equations and boundary conditions

### 2.2.1 Wave model

As already mentioned, the wave-related part of the model is based on NHWAVE. Its full description can be found in Ma et al. (2012, 2014); therefore, only a summary of the most important model features is given here. NHWAVE solves incompressible, nonhydrostatic Navier-Stokes equations in a three-dimensional domain, formulated in Cartesian horizontal coordinates and boundary-following vertical $\sigma$-coordinates, defined as:

$$\sigma = (z + H)/(H + \eta) = (z + H)/D, \tag{1}$$

for $z \in [-H(x), \eta(x,t)]$. In the $xz$-space, in which the present coupled ice–wave model is formulated, the governing equations are the mass and momentum conservation equations:

$$\frac{\partial D}{\partial t} + \frac{\partial (Du)}{\partial x} + \frac{\partial \omega}{\partial \sigma} = 0, \tag{2}$$

$$\frac{\partial (Du)}{\partial t} + \frac{\partial (Du^2 + \frac{1}{2}gD^2)}{\partial x} + \frac{\partial (Du\omega)}{\partial \sigma} = gD\frac{\partial H}{\partial x} - \frac{D}{\rho}\left(\frac{\partial p}{\partial x} + \frac{\partial p}{\partial \sigma}\frac{\partial \sigma}{\partial x}\right) + DS_{\tau_x}, \tag{3}$$

$$\frac{\partial (Dw)}{\partial t} + \frac{\partial (Duw)}{\partial x} + \frac{\partial (Dw\omega)}{\partial \sigma} = -\frac{1}{\rho}\frac{\partial p}{\partial \sigma} + DS_{\tau_z}, \tag{4}$$

where $g$ denotes acceleration due to gravity, $p$ – the dynamic pressure, $u$, $w$ are water velocity components in $x$ and $z$ direction, respectively, $\omega$ is the velocity component perpendicular to the $\sigma$-surfaces, and $(S_{\tau_x}, S_{\tau_z})$ are turbulent diffusion terms, assumed equal to zero in the present work. The free surface is obtained explicitly from the vertically-integrated continuity equation (2). To close the system of equations, (2)–(4) are supplemented by the Poisson equation for pressure (Ma et al., 2012; Orzech et al., 2016).

At the bottom, $z = -H$, the kinematic and free-slip boundary conditions for velocity, and the Neumann boundary condition for pressure are:

$$w = -u\frac{\partial H}{\partial x}, \tag{5}$$

$$\frac{\partial u}{\partial \sigma} = 0, \tag{6}$$

$$\frac{\partial p}{\partial \sigma} = -\rho D\frac{\mathrm{d}w}{\mathrm{d}t}. \tag{7}$$

Boundary conditions at the free surface, $z = \eta$, not covered with ice are:

$$w = \frac{\partial \eta}{\partial t} + u\frac{\partial \eta}{\partial x}, \tag{8}$$

$$\frac{\partial u}{\partial \sigma} = 0, \tag{9}$$

$$p = 0. \tag{10}$$

In the model applications presented in this work, sponge layers are applied at the left and right boundary, and waves are generated inside the model domain with a so-called internal-wavemaker technique, in which a source term is added to the model equations at the wave generation location, and the waves propagate out of this location in both directions (Ma et al., 2014).

### 2.2.2 Sea ice model

The sea-ice-related part of the model can be formulated as a set of the following ordinary differential equations:

$$\frac{d\theta_i}{dt} = \omega_i, \qquad i \in \mathcal{I}_{\mathrm{g}}, \tag{11}$$

$$\frac{dz_i}{dt} = w_i, \qquad i \in \mathcal{I}_{\mathrm{g}}, \tag{12}$$

$$I_{g,i}\frac{d\omega_i}{dt} = M_{wv,i} + M_{b,i} - M_{b,i-1} + l_i(F_{t,i} - F_{t,i-1}), \qquad i \in \mathcal{I}_{\mathrm{g}}, \tag{13}$$

$$m_i\frac{dw_i}{dt} = F_{wv,i} + F_{z,i} - F_{z,i-1}, \qquad i \in \mathcal{I}_{\mathrm{g}}, \tag{14}$$

$$\frac{dM_{b,i}}{dt} = -k_{n,i}I_{b,i}(\omega_i - \omega_{i+1}), \qquad i \in \mathcal{I}_{\mathrm{b}}, \tag{15}$$

$$\frac{dF_{t,i}}{dt} = k_{t,i}h_{b,i}v_{t,i}, \qquad i \in \mathcal{I}_{\mathrm{b}}, \tag{16}$$

$$\frac{dF_{z,i}}{dt} = k_{n,i}h_{b,i}v_{z,i}, \qquad i \in \mathcal{I}_{\mathrm{b}}. \tag{17}$$

Equations (11) and (12) are definitions of the angular and translational velocities of the grains, respectively. The angular-momentum equations (13) describe changes of $\omega_i$ due to moments of forces acting on the grains. Analogously, the linear-momentum equations (14) describe changes of the vertical velocity $w_i$ due to forces acting on the grains. The terms on the right-hand-side of (13) and (14) can be calculated from the remaining equations (15)–(17). As in all DEMs, the bonds transmit both torques and forces. Relevant in the present configuration are: bending moments $M_{b,i}$, resulting from the relative rotation (rolling) of the bonded grains in the $xz$ plane; torques $l_iF_{t,i}$ acting on the grain boundaries due to tangential forces resulting from translational shear displacement of the grains (with velocity $v_{t,i}$); and the vertical component of the sum of normal and tangential forces, $F_{z,i}$, resulting from relative displacement of the grains (with vertical velocity $v_{z,i}$). As can be seen, in (15)–(17) linear relationships between displacement and force are assumed, which is typical for DEM models, see Herman (2016) and, for a detailed algorithm for calculating the displacements and forces in a fully 3D case, Wang (2009) and Wang and Alonso-Marroquin (2009). Finally, the first terms on the right-hand-side of (13) and (14) denote the net moment of forces and the net vertical force, respectively, from the wave motion underneath the ice. They are calculated by integrating the contribution from waves over the wetted surface of the grains. Their detailed formulation is given further in Section 2.2.3.

Note that, in a general case, although the value of $F_{t,i}$ characterizes the bond connecting two neighboring grains, the torque related to this force acting on these grains would be different if $l_i \neq l_{i+1}$. Note also that the horizontal component of the normal and tangential forces would be relevant only for horizontal displacements of the grains, which are not taken into account here.

As noted earlier, all forces and moments are formulated for a unit width of grains and bonds.

The stresses acting on bonds are calculated according to the classical beam theory, so that:

$$\tau_i \quad = \quad \frac{|F_{t,i}|}{h_{b,i}}, \qquad i \in \mathcal{I}_{\mathrm{b}}, \tag{18}$$

$$\sigma_{c,i} \quad = \quad \frac{F_{n,i}}{h_{b,i}} + \frac{|M_{b,i}|h_{b,i}}{I_{b,i}}, \qquad i \in \mathcal{I}_{\mathrm{b}}, \tag{19}$$

$$\sigma_{t,i} \quad = \quad -\frac{F_{n,i}}{h_{b,i}} + \frac{|M_{b,i}|h_{b,i}}{I_{b,i}}, \qquad i \in \mathcal{I}_{\mathrm{b}}, \tag{20}$$

where $F_{n,i}$ denotes the normal force (i.e., along the bond length). The stresses are evaluated for every bond at every model time step. If at least one of the three stress components exceeds the bond strength, i.e., if $\tau_i > \tau_{br,i}$ or $\sigma_{c,i} > \sigma_{c,br,i}$ or $\sigma_{t,i} > \sigma_{t,br,i}$, the bond breaks. In bonded-particle models this is typically achieved by instantaneously setting the Young's modulus, as well as the forces and moments transmitted by this bond, to zero. This approach, based on an assumption that breaking happens infinitely fast, is well known to produce too brittle behavior, unrealistic in many materials. Some models therefore introduce a softening mechanism, ensuring that stress in broken bonds drops gradually instead of instantaneously (see, e.g., Lisjak and Grasselli, 2014). In the present model, breaking is extended in time by assuming that stresses acting on a bond that undergoes breaking drop to zero gradually over a certain time $t_{br}$. Numerical tests showed that $t_{br} \sim 0.1$ s is enough to remove spurious effects associated with instantaneous breaking. The influence of $t_{br}$ on the model behavior is demonstrated in Section 3.3.

### 2.2.3 Sea ice–wave coupling

In the present model, the discretization of the model domain in the vertical direction is modified so that a prescribed number $N_{l,ice}$ out the total of $N_l$ layers is used to accommodate the ice (Fig. 1). That is, the uppermost $N_{l,ice}$ layers have a constant thickness equal to $h_f/N_{l,ice}$, where $h_f$ denotes the draft of the ice. The remaining $N_l - N_{l,ice}$ layers are divided uniformly from the bottom, $z = -H(x)$ to $z = \eta(x,t) - h_f$. Thus, the thickness of the upper model layers does not vary in time and at each time step the ice grains' boundaries coincide with boundaries of the cells of the wave model. This fact significantly simplifies the formulation of boundary conditions along the horizontal and vertical ice surfaces. At the lower surface of the ice we have:

$$w \quad = \quad w_i, \tag{21}$$

$$\frac{\partial u}{\partial \sigma} \quad = \quad 0, \tag{22}$$

$$\frac{1}{D}\frac{\partial p}{\partial \sigma} \quad = \quad -\rho\frac{\mathrm{d}w_i}{\mathrm{d}t}. \tag{23}$$

Analogously, at the vertical ice surfaces:

$$u \quad = \quad u_i, \tag{24}$$

$$\frac{\partial w}{\partial x} \quad = \quad 0, \tag{25}$$

$$\frac{\partial p}{\partial x} \quad = \quad -\rho\frac{\partial u_i}{\partial t}. \tag{26}$$

(Note that $u_i = 0$ in the present model version.) As can be seen, a free-slip condition is assumed for velocity components tangential to the ice surface.

In the immersed-boundary method, the influence of the ice on the surrounding water is taken into account by adding an additional forcing term $F_{ice}$ to the momentum equations at the second step of the two-step second-order Runge-Kutta scheme, used in NHWAVE to numerically integrate the governing equations (Ha et al., 2014; Ma et al., 2016). By definition, $F_{ice} \neq 0$ only along the boundaries of floating/submerged objects (points marked with red crosses in Fig. 1). Details of the formulation of this force can be found in Ha et al. (2014) and in references cited there. Linear interpolation of velocities close to ice boundaries is used, as recommended by Fadlun et al. (2000) and Ha et al. (2014).

To close the wave–ice interaction problem, the forcing from water to the ice has to be passed to the ice model. This forcing can be obtained by integrating the dynamic pressure $p$ over the surface area of an submerged object. Due to the specific geometry and assumptions described in previous sections, the formulation of this forcing is relatively straightforward. As the horizontal motion of the grains is not taken into account and the tilt of the grains is likely to remain small (so that $\sin\theta_i$ is close to zero and $\cos\theta_i$ close to one), contribution of pressure force and momentum acting on the vertical surfaces of end grains can be omitted. Thus, the moment $M_{wv,i}$ used in (13), and the vertical component of the wave-induced force $F_{wv,i}$ in (14) are:

$$M_{wv,i} = \int_{x_i-l_i}^{x_i+l_i} p(l)\mathbf{n}_i \times \mathbf{r}_i dl, \qquad i \in \mathcal{I}_g, \tag{27}$$

$$F_{wv,i} = \cos\theta_i \int_{x_i-l_i}^{x_i+l_i} p(l)dl, \qquad i \in \mathcal{I}_g, \tag{28}$$

where $l$ denotes distance along the lower grain surface, $\mathbf{n}_i = [-\sin\theta_i, \cos\theta_i]$ is a unit vector normal to that surface, and $\mathbf{r}_i$ is a vector of length $l$ tangential to it. Assuming linear variability of pressure between $p_{i-1}$ and $p_i$, as well as between $p_i$ and $p_{i+1}$, it is straightforward to evaluate the integrals in (27) and (28) to obtain:

$$M_{wv,i} = \frac{l_i^3}{3\Delta x}(p_{i+1} - p_{i-1}), \qquad i = 1,\dots,N_g, \tag{29}$$

$$F_{wv,i} = 2l_i\left[p_i + \frac{2l_i}{8\Delta x}(p_{i+1} - 2p_i + p_{i-1})\right]\cos\theta_i, \qquad i = 1,\dots,N_g. \tag{30}$$

## 2.3 Numerical implementation

The code of the sea ice model is written as an additional module included in NHWAVE. A simplified flowchart of the coupled model is shown in Fig. 3. Due to more strict stability requirements of the sea ice part of the model, it is solved with a shorter time step $\Delta t_{\text{ice}} = \gamma_t \Delta t_{\text{wave}}$, with $\gamma_t < 1$. In simulations presented in this paper, $\gamma_t = 1/150$ was used. The time step of the ice model is limited by the grain size used and by mechanical ice properties, with more stiff ice (higher $E_b$) requiring smaller $\Delta t_{\text{ice}}$.

## 3 Results

### 3.1 Model setup

In this section, the model is applied to a series of simulations in which a single ice floe with a given thickness $h_i$ and length $L_o$ is moving on waves with a given open-water wavelength $L_{w,0}$. A summary of the model setting is given in Table 1. The water depth is constant $H = 10$ m, and the water column is divided into $N_l = 30$ layers. The number of "ice layers" $N_{l,ice}$ depends on the ice thickness, but is never lower than 3. The horizontal resolution of the model, i.e., the cell size of the wave model $\Delta x$ and the horizontal dimensions of the grains $2l_i$, equals 0.5 m. Preliminary simulations with the standalone NHWAVE model were performed to verify whether $\Delta x$ is sufficiently small and $N_l$ sufficiently large to reproduce the shortest waves considered with satisfactory accuracy. The results showed that for the whole range of wavelengths analyzed, no significant loss of energy during propagation was observed. Thus, the attenuation present in the results described further in Sections 3.2 and 3.3 originates in the sea ice module: due to damping in the bonds which are not perfectly elastic. This undoubtedly is an undesired property of the numerical scheme used in the sea ice module; however, it has been shown in tests with artificially modified damping in bonds that it does not influence the results in terms of the floe sizes obtained – see Section 4 for a discussion.

It is also worth stressing that – as in all DEM models – in simulations that are designed to reproduce the behavior and macro-properties of any particular specimen of a brittle material (its strength, elastic modulus, and so on), the microscopic properties of grains and bonds have to be carefully calibrated (see, e.g., Potyondy and Cundall, 2004; Koyama and Jing, 2007, for examples how the coefficients of the bond and contact models are calibrated in order to take into account their dependence on grain size). As the results presented here are not calibrated to any real-world case, this issue is not further investigated. For realistic applications of the model, its parameters ($\lambda$, $\lambda_{ns}$, $E_b$, $\sigma_{t,br}$ and so on) can be adjusted to obtain desired macroscopic sea ice properties.

In the simulations described in Sections 3.2 and 3.3, a number of combinations of $h_i$, $L_o$ and $L_{w,0}$ are considered, with the range of values 0.3–3.0 m, 5–500 m and 25–84 m, respectively. For $H = 10$ m, the range of $L_{w,0}$ corresponds to wave periods between 4.04 and 9.19 s and to $kH$ values between 2.5 and 0.75 (where $k$ denotes the wave number). The thickness of both grains and bonds is identical.

The simulations were performed first without ice breaking in order to analyze the spatiotemporal variability of stress in the ice, as described in Section 3.2. Subsequently, the bonds' strength was reduced to a number of values to study ice breaking pattern, analyzed in Section 3.3.

### 3.2 Stress variability in continuous ice

During the motion of the modelled ice floe on waves, the bonds undergo tensile, compressive and shear stress related to the relative displacement and rotation of neighboring grains. In the simulations described here, the compressive and tensile stresses had comparable amplitudes, whereas the shear stress was two–three orders of magnitude lower. All bond breaking events in simulations from Section 3.3 happened due to tensile failure and therefore $\sigma_{t,i}$ is analyzed here as the most relevant stress component.

Figure 4 shows the vertical displacement of the ice and the tensile stress acting on bonds in function of time and distance from the ice edge. As can be seen in the diagrams, the amplitude of stress acting on bonds increases from zero at the ice edge (where the amplitude of $z_i$ is largest) towards a maximum value $\sigma_{t,\max}$ at a certain distance from the ice edge (see the pink dot in the lower plot in Fig. 4b). Figure 5a,c shows the value of $\sigma_{t,\max}$ for different combinations of ice thickness and floe lengths; the location of the stress maximum (measured relative to the ice edge) is shown in Fig. 5b,d. For a given ice thickness, the value of $\sigma_{t,\max}$ increases with increasing floe size, as the floes' response changes from rigid motion (very small floes) to flexural motion (larger floes). Up from a certain floe size, equal to between one and two wavelengths, no further increase of $\sigma_{t,\max}$ is observed, i.e., the stress saturates to a value specific for a given ice thickness. For a given floe length, the influence of ice thickness on $\sigma_{t,\max}$ is less trivial: there is a certain value of $h_i$ for which $\sigma_{t,\max}$ reaches the highest value, and for larger floes this maximum (Fig. 5a) shifts towards thicker ice. The reason for the drop of stress in very thick ice is that a lot of wave energy is reflected at the ice edge, leading to lower amplitudes within the ice itself. Moreover, thick floes are more rigid, with reduced strain and thus lower stress levels. For very small floes, $\sigma_{t,\max}$ occurs in the middle of the floe and thus its location is ice-thickness independent; for larger floes, location of $\sigma_{t,max}$ moves further from the ice edge with increasing ice thickness (Fig. 5d). For a given ice thickness, location of $\sigma_{t,\max}$ moves away from the ice edge with increasing floe size (Fig. 5b).

Apart from the ice properties, the value and location of $\sigma_{t,\max}$ are influenced by the characteristics of the incoming waves, as shown in Fig. 6 for two selected ice thicknesses and for a range of floe lengths. For a given open-water wavelength $L_{w,0}$, $\sigma_{t,\max}$ increases with increasing floe length up to a certain "saturation" value (Fig. 6a,c). On the other hand, for large floes there's a certain open-water wavelength producing maximum tensile stress (assuming the same incident wave amplitude). Again, this is related to both wave reflection at the ice edge and the response of the ice itself. For very short waves, strong reflection leads to lower wave amplitude within the ice; for very long waves, on the other hand, reflection and damping within the ice are weaker, but the wave steepness is small as well, leading to less intense flexural motion of the ice (see also Montiel et al., 2013). Most importantly, the location of $\sigma_{t,\max}$ is almost independent of the incoming wavelength (Fig. 6b,d; note that the size of the grains, and thus the effective resolution of the model, equals 0.5 m, so that the differences seen in the figures, especially in the case of $h_i = 0.5$ m, amount to just two–three grains).

For large floes, a few stress maxima with decreasing amplitude can be observed behind the main one, as shown in Fig. 7. Sufficiently far from the floe edge, the stress amplitude decreases gradually, depending on the damping rate (which depends on ice thickness and wave characteristics; see also Fig. 4 and Section 3.1 for the discussion on the sources of damping in the present model version). At the rear side of large floes, small-amplitude ripples are observed before the stress drops to zero – similar increase of the amplitude of the vertical motion of elastic plates at their downwave ends has been observed and modelled, e.g., by Kohout et al. (2007) and Yoon et al. (2014). As already mentioned, small floes ($L_o < L_{w,0}/2$) have only one stress maximum, as they undergo bending around their symmetry axis (Fig. 7b). As the floe size exceeds $L_{w,0}/2$, the symmetry gradually vanishes and the second maximum appears when $L_o$ is close to $L_{w,0}$.

### 3.3 Breaking of uniform ice by regular waves

The spatiotemporal variability of tensile stress in the ice, described above, is crucial for the evolution of ice breaking and the resulting floe-size distribution. Figure 8 illustrates how breaking of a large floe ($L_o = 500$ m) progresses from the ice edge deeper and deeper into the ice, producing small floes with lengths comparable to the distance of $\sigma_{t,\mathrm{max}}$ to the ice edge. An individual wave is "responsible" for a few breaking events (between one and three in the case shown in Fig. 8; up to five in other analyzed cases) and thus produces a few new ice floes. In thinner ice, the number of new cracks per wave period tends to be larger, i.e., breaking progresses into the ice faster than in stronger, thicker ice. Moreover, as can be expected, the final width of the zone of broken ice is ice-strength dependent as well and, in the cases analyzed, increases roughly linearly with decreasing bond strength (not shown). The resulting breaking pattern is not perfectly regular, but the floe-size distribution is very narrow. In the simulation presented in Fig. 8, in which the distance of $\sigma_{t,\mathrm{max}}$ from the ice edge equaled 8 m (yellow curve in Fig. 6b), only four floe sizes were obtained, 6.5, 7.0, 7.5 and 8.0 m, with the mode of the distribution at 7.0 m. Generally, the location of $\sigma_{t,\mathrm{max}}$ appears to constitute an upper bound on the size of floes detached from the edge of continuous ice, and breaking takes place not farther than a few grains in front of that limiting location.

Once the small floes break off the receding ice edge, they begin to move as almost-rigid bodies, changing their vertical position and rotating around their symmetry axis (Fig. 9). In the present model, in which the horizontal component of ice motion is not included, neighboring grains do not interact with each other if they are not bonded. Thus, a very important mechanism of wave-energy attenuation is not taken into account: floe–floe collisions. Consequently, the model produces lower attenuation rates in broken ice than in the initial continuous ice sheet (Fig. 9b). This behavior is fully consistent with the model assumptions, but not realistic. As a result, the width of the zone of broken ice is likely overestimated in the present model version. However, this drawback hardly influences the overall breaking patterns, as they are very robust to changes of the model configuration. As an example, Figure 10 shows the results of a simulation analogous to that presented in Fig. 8, but with incoming waves with a Jonswap energy spectrum (one of widely used idealized models of wave energy spectra, suitable for a wide range of wind and fetch conditions). As can be seen, even though the waves are irregular and breaking takes places in short episodes (associated with wave groups) separated by quieter periods without formation of new cracks, the final floe-size distribution is as regular as that produced by sine waves. Another important mechanism not taken into account in the present version of the model is multiple wave scattering by small ice floes detached from the ice edge. As Montiel and Squire (2017) have recently shown, scattering may lead to both destructive and constructive interference, thus contributing to local decrease or increase of the wave amplitude and strain of the ice. The net effects of these processes on the wave attenuation rates and ice breaking patterns are hard to estimate and presumably sensitive to the details of any particular configuration. (Note that the present model is capable of simulating multiple scattering, but not in the configuration used here, in which the grains of the sea ice module occupy full cells of the wave module, so that no water–ice boundary conditions are applied at the vertical walls of neighboring grains.)

Finally, it is worth noticing that the regular floe pattern described above is obtained only in simulations in which the "delayed" bond breaking mechanism, described at the end of Section 2.2.2, was activated. Figure 11 compares the results of two

similar simulations, one with instantaneous and one with "delayed" bond breaking. If breaking is instantaneous, sudden drop to zero of all stress components at the broken location produces short-wave disturbance propagating out of this location in both directions (Fig. 11b). The excess stress related to that disturbance, combined with stress induced by the propagating wave, leads to rapid bond breaking in neighborhood of the initial breakage, producing very small ice floes, typically 2–3 grains in size

(compare Fig. 11a to Fig. 8b). If, to the contrary, the drop of stress during bond breaking is extended over a time period of just less than 0.1 s, it is sufficient to suppress the amplitude of the breaking-induced disturbance to insignificant levels (Fig. 11c). Consequently, no additional breaking takes place around the initial crack.

## 4   Discussion and conclusions

In this paper, a coupled wave–ice model was used to analyze wave-induced stress in sea ice and the resulting patterns of sea ice

breaking. The most important results can be summarized as follows: (i) breaking of a continuous ice sheet by waves produces floes of almost equal sizes, dependent on the thickness/strength of the ice, but not on the characteristics of the incoming waves; (ii) this breaking pattern results from the fact that maximum tensile stress experienced by the ice is located at a distance from the ice edge that does not depend on incoming wavelength; (iii) the incoming wave characteristics, together with ice properties, decide upon the value of the maximum stress, thus deciding whether breaking takes place or the ice remains intact; (iv) for a

given floe size, there exist ice thickness and incident wave length for which the stress reaches maximum and thus breaking is most likely to occur.

   As no attempt at calibrating the model against observational data was made, the numbers obtained as a result of the simulations might be unrealistic. Also, as has been already mentioned in the previous section, there are a number of mechanisms of wave-energy dissipation that are not included in the present version of the model (floe–floe collisions, ice–water friction,

multiple scattering by the floes already broken off the ice edge, etc.). However, these facts do not affect the general conclusions formulated above. The present results agree with the findings of Squire et al. (1995), described in the introduction, and provide another evidence – obtained with a very different model than that of Squire and colleagues – in favor of the hypothesis that it is the ice itself (its thickness and strength) and not the incident waves that decide upon the dominating floe size in MIZ, at least during the initial stages of ice breaking (at later stages, many other factors lead to further fragmentation of ice floes, producing

wide, heavy-tailed floe-size distributions typically observed in inner parts of MIZ; see, e.g., Toyota et al., 2011, 2016, and references there). In particular, it is worth stressing that in terms of the floe size resulting from breaking, the results are not sensitive to the modelled attenuation rates of wave energy (which, as already mentioned in Section 3.1, has been demonstrated in model runs with artificially modified damping in bonds connecting grains). Breaking takes place within a narrow zone of enhanced strain close to the edge of the yet unbroken ice. Again, this is consistent with the observational and modelling results

of Squire (1984b); Squire et al. (1995), who found that breaking is likely only within a region where the secondary ice-coupled waves contribute to the increased vertical deflection and thus strain of the ice. As the amplitude of these waves decays very fast with the distance from the ice edge, so does the probability of breaking, independently of the attenuation coefficient of the gradually decaying propagating wave.

If further research confirms these results, it will have important consequences for formulating parameterizations of wave–ice interactions for large-scale sea ice models, so that the information on incoming waves (especially wave steepness) is used to determine whether breaking of ice takes place, but the maximum floe size $L_{\max}$ is estimated based on ice properties themselves. (Note that, as already mentioned in the introduction, in most parameterizations $L_{\max}$ is the only variable parameter describing the FSD; the shape of the FSD for $L_o < L_{\max}$ is assumed to be a power law with a prescribed exponent. Note also that besides bending, a number of other wave-related processes may contribute to floe breaking and thus to shaping the FSD, including floe–floe collisions, overwash, rafting, etc. These processes are dependent on wave steepness, and thus amplitude; presumably, they modify the slope of the FSD, although no observational data exist that would allow to formulate this dependence as a functional relationship.)

The model presented in this paper is undergoing further development as part of a research project currently in progress. In the new version, horizontal ice motion and ice contact mechanics will be implemented (by adapting algorithms from the DESIgn model; see Herman, 2016), enabling to run the model to study floe–floe collisions and situations with significant drift and/or surge motion of ice. At later stages of the project, it is planned to extend the model to two horizontal dimensions (the NHWAVE model is three-dimensional, and significant parts of the sea ice module have already been coded for two horizontal dimensions as well). This will make it possible to analyze how the directional width of the energy spectra of incoming waves, as well as the angle between the wave propagation direction and the ice edge affect the results obtained in this study. It is also worth noticing that the code of the model can be easily extended by, e.g., replacing the free-slip boundary conditions for velocity at the wetted surface of the ice with other types of boundary conditions, or by including wind or other processes already implemented in NHWAVE. It should be stressed that NHWAVE is a very general hydrodynamic model that can be applied to a wide range of conditions: it does not make any assumptions regarding the irrotationality of the flow (as many sea ice–wave interaction models do) or the type of the wave forcing. Although in the computations presented in this paper the water depth was relatively shallow ($H = 10$ m), deep-water waves can be simulated without significant increase in computational costs, because the model enables non-equally spaced $\sigma$-layers, with thickness adjusted to the vertical structure of the wave. The model also accepts a number of types of boundary conditions, handles drying and flooding of grid cells, etc. All these functionalities can be used in coupled wave–ice simulations, making it a very flexible tool suitable for a wide range of conditions. A serious limitation, however, are very high computational costs of this modelling approach. This makes the model suitable for analyzing details of selected processes – like in this paper – rather than for practically-oriented applications in sea ice and wave hindcasting and forecasting.

*Author contributions.* A. Herman designed and implemented the model, planned and performed the simulations, analyzed the results, and wrote the text.

*Acknowledgements.* This work has been supported by the Polish National Science Centre research grant No. 2015/19/B/ST10/01568 ("Discrete-element sea ice modeling – development of theoretical and numerical methods"). I am very grateful to Fabien Montiel, Anton Kulchitsky and the anonymous reviewer for their constructive criticism and insightful comments that helped to improve the quality of this paper.

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

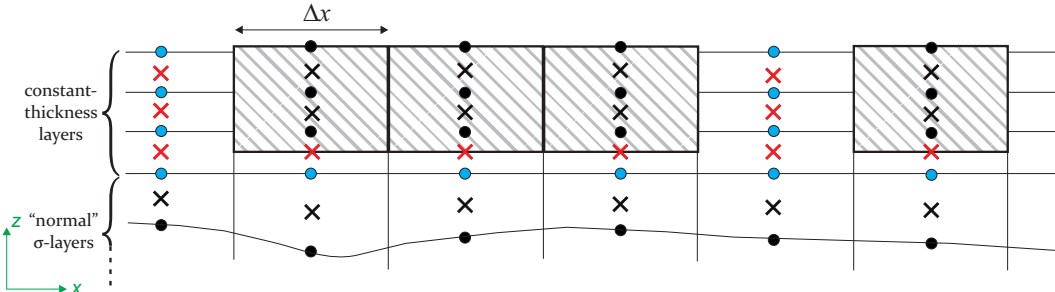

**Figure 1.** Sketch of the grid organization and spatial arrangement of variables in the coupled wave–ice model, for the case of three constant-thickness uppermost layers ($N_{l,ice} = 3$) accommodating the ice 'grains' (dashed boxes). Crosses denote velocity points, dots – pressure points. Locations in which the immersed-boundary forcing is applied are shown in red, pressure points affected by the boundary – in blue (note that, in accordance with the immersed boundary method, the model equations are solved everywhere inside the model domain, independently of ice being present in a given grid cell or not). See text for more details.

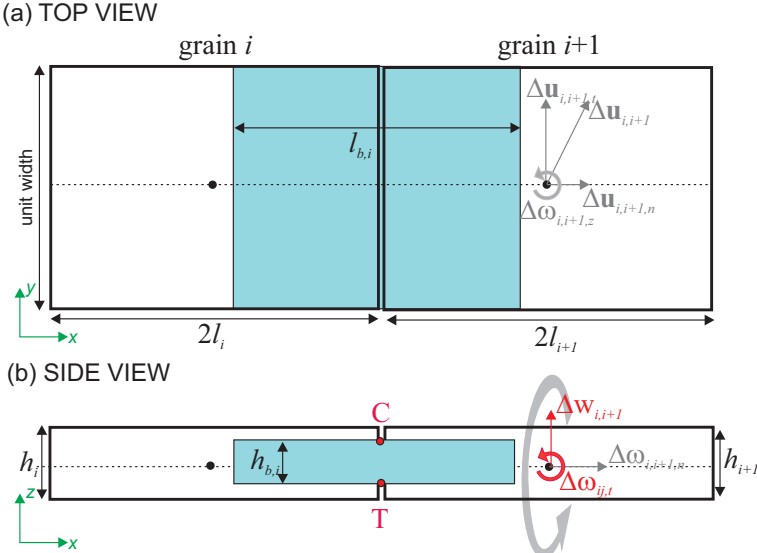

**Figure 2.** Top view (a) and side view (b) of two neighboring grains (white rectangles) connected with a bond (blue rectangle). Relative translational and angular velocity differences relevant for the present study are shown in red (rotation in the $xz$ plane and vertical displacement), the remaining velocity components – in gray. Red dots with labels 'C' and 'T' in (b) mark the locations of maximum compressive and tensile stress, respectively, acting on the bond if the relative rotation is directed as shown in red.

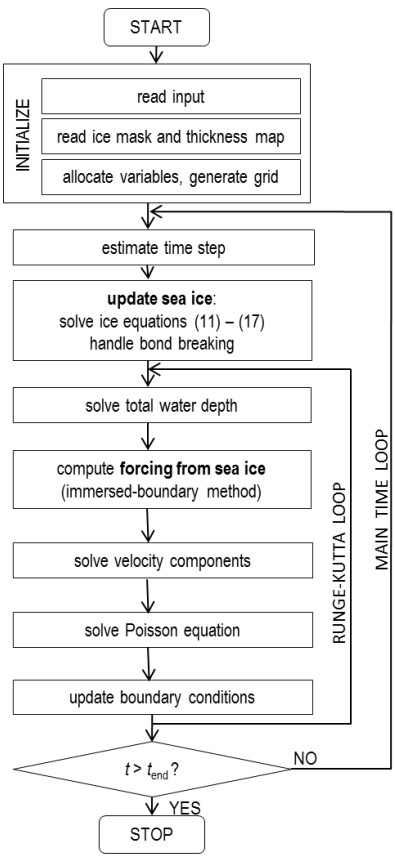

**Figure 3.** Simplified flowchart of the coupled wave–ice model.

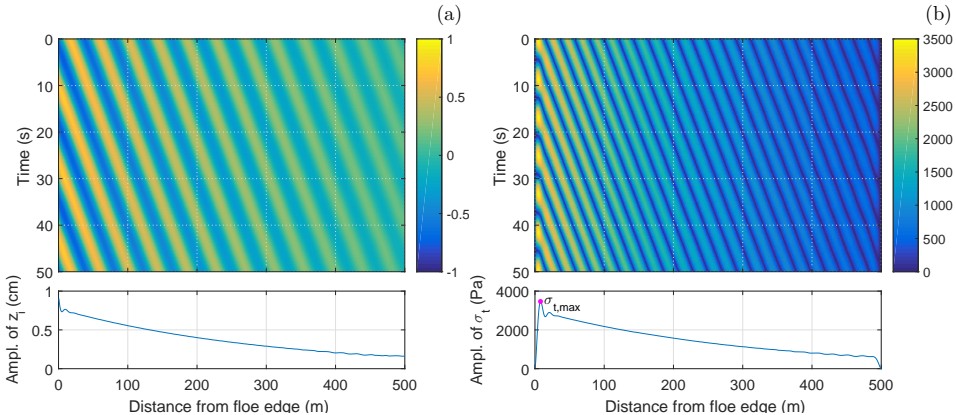

**Figure 4.** Simulated space–time variability of the ice vertical displacement $z_i$ (a; in cm) and the tensile stress $\sigma_t$ (b; in Pa) for an ice floe with length $L_o = 500$ m; ice thickness $h_i = 0.5$ m, open-water wavelength $L_{w,0} = 42$ m. Lower diagrams show the amplitude of $z_i$ and $\sigma_t$ in function of the distance from the ice edge. Magenta dot in (b) marks $\sigma_{t,\mathrm{max}}$.

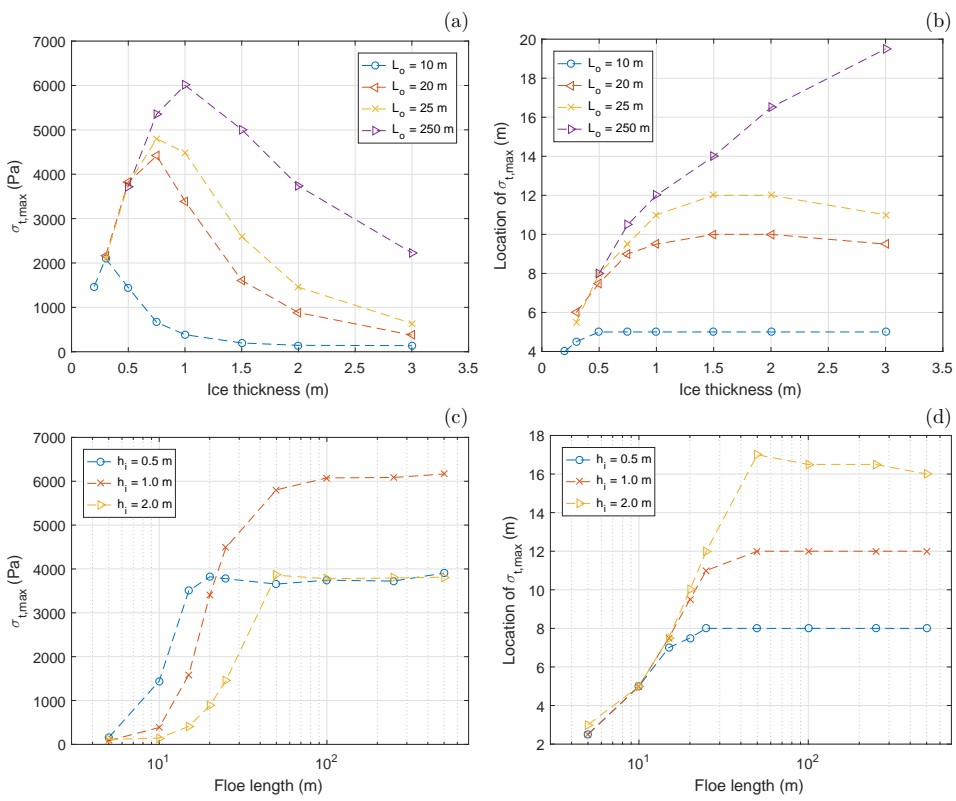

**Figure 5.** Simulated maximum tensile stress $\sigma_{t,\max}$ (a,c) and location (distance from the up-wave ice edge) at which it occurs (b,d) for different ice thickness $h_i$ and floe length $L_o$ values; open-water wavelength $L_{w,0} = 42$ m. Note that the $x$-axis in (c,d) is logarithmic.

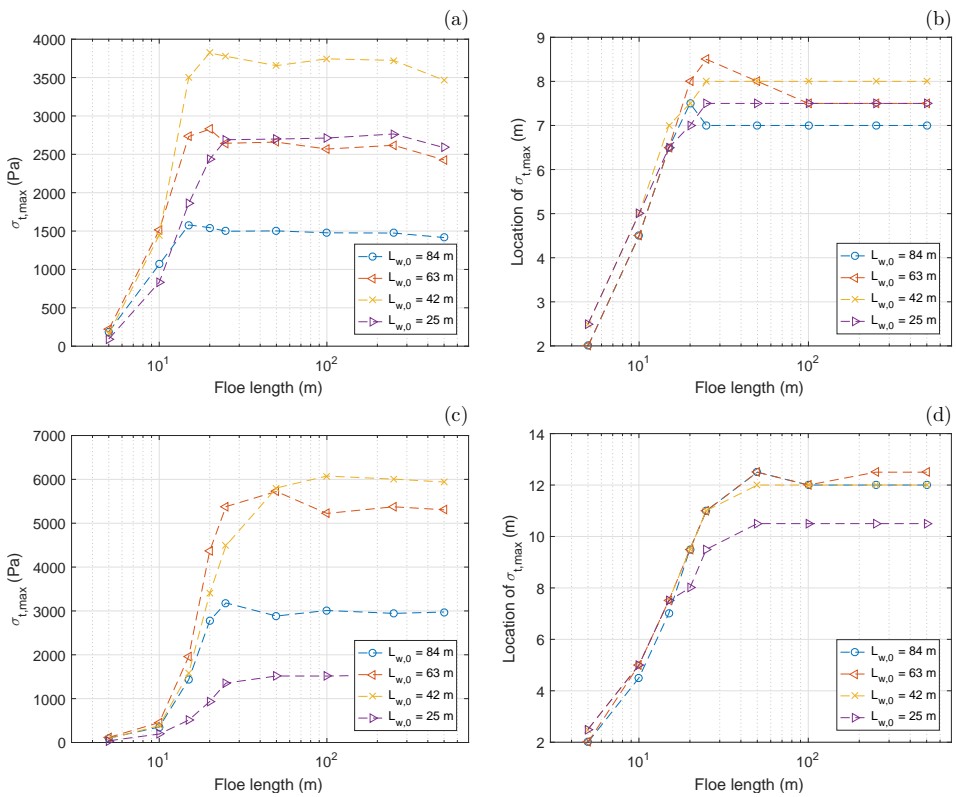

**Figure 6.** Simulated maximum tensile stress $\sigma_{t,\max}$ (a,c) and location (distance from the upwave ice edge) at which it occurs (b,d) for different floe length $L_o$ and open-water wavelength $L_{w,0}$ values; ice thickness $h_i = 0.5$ m (a,b) and $h_i = 1.0$ m (c,d). Note that the $x$-axis is logarithmic.

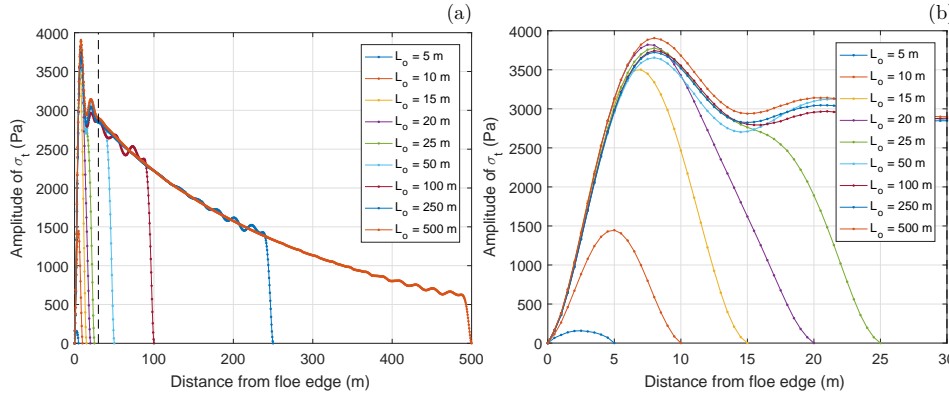

**Figure 7.** Amplitude of the tensile stress $\sigma_t$ in function of the distance from the upwave ice edge for different floe length $L_o$; open-water wavelength $L_{w,0} = 42$ m, ice thickness $h_i = 0.5$ m. The plot in (b) is a close-up of the region to the left of the dashed line in (a).

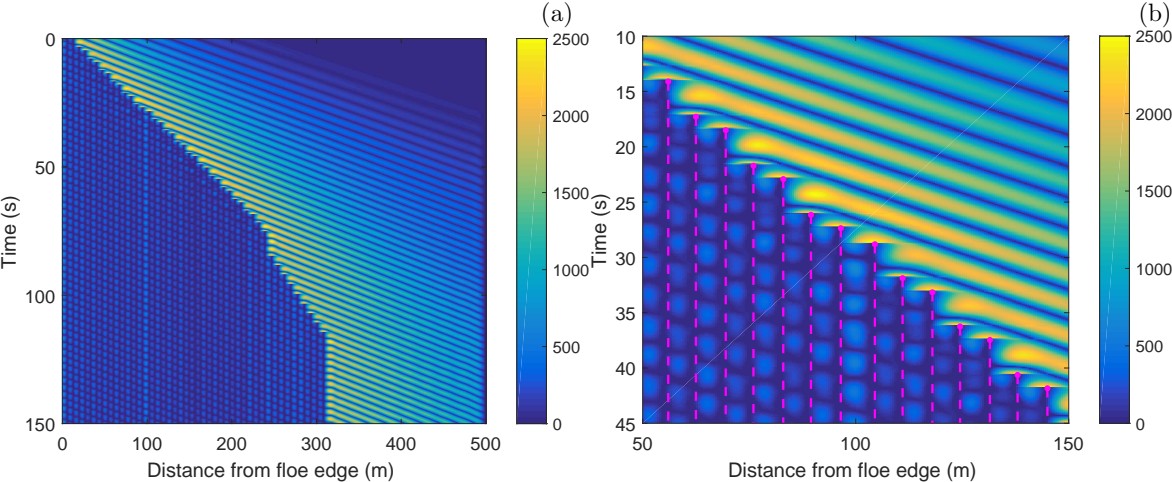

**Figure 8.** Simulated space–time variability of the tensile stress $\sigma_t$ (Pa) for an ice floe with length $L_o = 500$ m undergoing progressive breaking; ice thickness $h_i = 0.5$ m, open-water wavelength $L_{w,0} = 42$ m, bond strength 2500 Pa. The plot in (b) is a subset of that in (a); breaking events are marked with magenta dots, broken bonds with dashed lines.

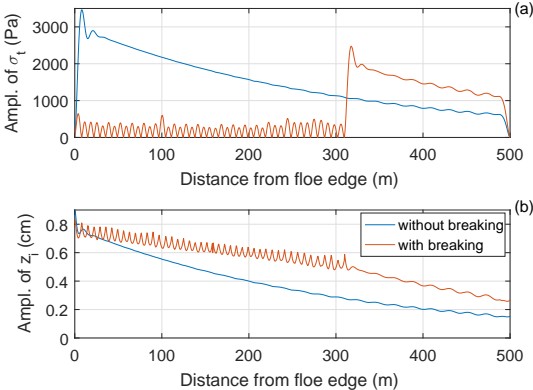

**Figure 9.** Amplitude of the tensile stress $\sigma_t$ (a) and vertical ice displacement (b) in simulations without and with ice breaking. Floe length $L_o = 500$ m, ice thickness $h_i = 0.5$ m, open-water wavelength $L_{w,0} = 42$ m, bond strength 2500 Pa.

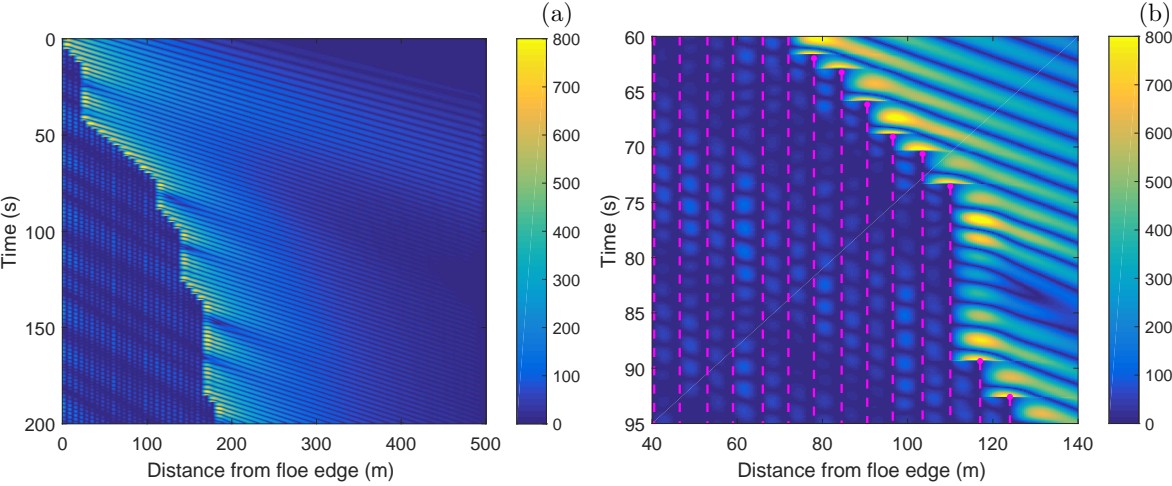

**Figure 10.** As in Fig. 8, but for irregular incoming waves with Jonswap energy spectrum (wave height and peak period corresponding to those of sine waves used in simulation from Fig. 8).

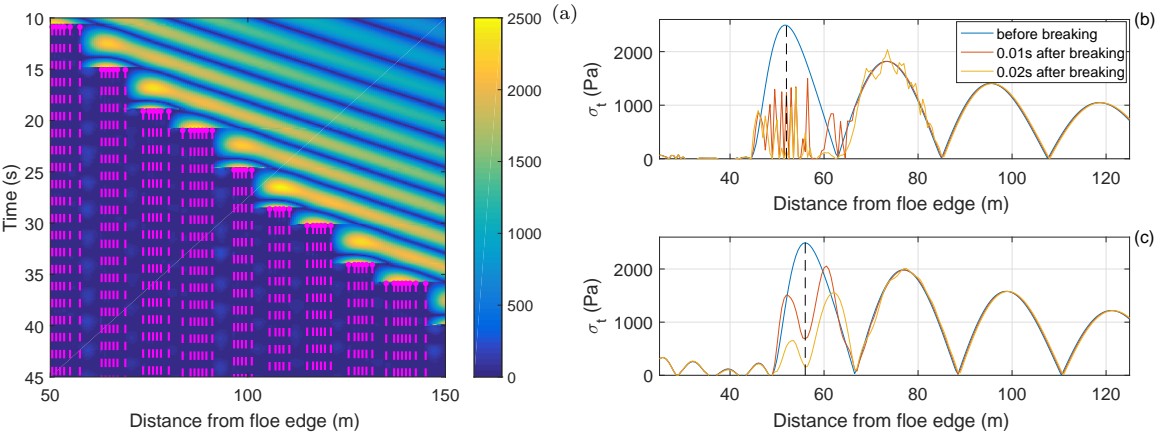

**Figure 11.** Comparison of the model behavior in simulations with instantaneous and "delayed" bond breaking: space–time variability of the tensile stress $\sigma_t$ (Pa) in a simulation analogous to that shown in Fig. 8, but with instantaneous bond breaking (a); and details of $\sigma_t$ in vicinity of a selected breaking event from a simulation with instantaneous (b) and "delayed" (c) bond breaking. The curves in (b,c) show $\sigma_t$ along a selected fragment of the ice floe before (blue) and shortly after (red and yellow) breaking, dashed black lines mark the location where breaking took place.

**Table 1.** Model parameters used in the simulations in Section 3.

| Variable | Value |
| --- | --- |
| Constant parameters: | |
| Water depth $H$ | 10 m |
| Basin length | 1500 m |
| Horizontal grid size $\Delta x$ | 0.5 m |
| Number of $\sigma$-layers $N_l$ | 30 |
| Number of "ice layers" $N_{l,ice}$ | $\max\{3, 3h_i\}$ |
| Width of sponge layers | 125 m |
| Internal-wavemaker location | 290 m |
| Bond length parameter $\lambda$ | 0.5 |
| Normal to shear stiffness ratio $\lambda_{ns}$ | 1.5 |
| Young's modulus $E_b$ | $1.0 \cdot 10^9$ Pa |
| Time step ratio $\gamma_t$ | 150 |
| Wave amplitude $a$ | 0.025 m |
| Variable parameters: | |
| Floe length $L_o$ | 5–500 m |
| Ice thickness $h_i$ | 0.3–3.0 m |
| Open-water wavelength $L_{w,0}$ | 25–84 m |
| Bond tensile strength $\sigma_{t,br}$ | 1500–3000 Pa |
| | ($\infty$ in simulations without breaking) |