# Peer review of "Wave-induced stress and breaking of sea ice in a coupled hydrodynamic-discrete-element wave-ice model"

_The Cryosphere, 2017_

## Referee Comment (RC1) · F. Montiel (Referee) · 22 Jun 2017

The article describes a new two-dimensional time-dependent model of ocean waves interactions with sea ice. The model arises from coupling the incompressible, nonhydro-static Navier-Stokes equations solver NHWAVE for the water domain with a discrete-element model (DEM) for the sea ice. The NHWAVE model is adapted from its original version to accommodate an ice cover with draught instead of a free surface. The DEM describes the ice cover as a discrete set of adjacent identical rectangular rigid bodies, referred to as grains, connected horizontally to each other by elastic beams, referred to as bonds, with same thickness as the grains. A coupled set of ordinary differential equations is derived for the angular and vertical velocity components of the grains, which accounts for the hydrodynamic forcing. The coupled water/ice equations are solved numerically with a second order Runge-Kutta time-stepping method. The model is used to investigate the response of a single floating ice floe with constant thickness under regular and irregular wave forcing, with the goal of analysing (i) the evolution of the stress field experienced by the floe and (ii) the floe size pattern obtained after repeated breaking of the floe.

I think this is an interesting and strong piece of work which addresses an important open scientific question. The manuscript is well written and the assumptions are clearly justified. The model described by the author is novel and the conclusions reached follow a reasonable interpretation of the results. My main concern is that the article lacks a detailed discussion on the advantages and drawbacks of this new modelling approach compared to existing waves/ice interactions models. For instance, it would be good to have an idea of the computational complexity of the model compared to others and the main parametric restrictions affecting this complexity, e.g. small water depth. I think it would have been nice to conduct a comparative study of wave propagation through an ice floe with an existing model for validation, but I understand this is not what this paper is about.

I support publication of the manuscript in *The Cryosphere* provided the authors address my comments below.

**General comments**

1. I think the Introduction section needs some work. To be more specific, I think it could be re-organised more efficiently so it is more relevant to the scientific question addressed in the rest of the manuscript. The first paragraph discusses (i) the geophysical context, i.e. Arctic sea ice reduction and the impact of ocean waves,

(ii) the parametrisation of wave/ice interactions in large scale forecasting models, (iii) the limited observational data, (iv) the attenuation rate of wave height in the MIZ and (v) the impact of waves on sea ice in the Southern Ocean. These are all relevant discussions, but they each need to be expanded and separated in different paragraphs. The second paragraph, on the other hand, lists the range of topics and associated publications on wave/ice modelling in the literature, even though it is not clear how they are relevant (if at all) to the present manuscript. I would suggest a more fit-to-purpose approach to the Introduction, in which the authors state their goals early on, and then discuss what has already been done to address these questions and how the new model proposed here goes beyond what is already established. I also suggest that the two paragraphs at the beginning of section 2, and before the start of section 2.1, are integrated as part of the Introduction, as they give a broad presentation of the two models used in this study, which is lacking in the current version of the Introduction. Additional comments on specific parts of the Introduction are given below.

2. As mentioned earlier, the manuscript needs to discuss in more depth other modelling studies on wave-induced breakup in the MIZ, particularly the papers referenced in page 2 (line 2), to which should be added the recent study by Bennetts et al. (2017) in *The Cryosphere*, in which the breakup is treated in a very different manner to that of the present model. More focus should then be added on how the present model differs from these existing models and what are its relative strengths and weaknesses.

3. The model described in section 2 is presented with much generality, and by doing so, the author introduced many variables that are later either neglected or set to be constant. I understand that the author wants to convey the generality of the modelling approach described here, but considering the main goals of the present study is to analyse the tensile stress in the bonds and breaking of a single floe with uniform properties, I do not think it is necessary to overcomplicate

the equations. More specifically, the turbulent diffusion terms in equations (3) and (4) are not needed and the quantities associated with the bonds and grains that are constant, e.g. thickness, width, mass, moment of inertia and elasticity parameters, do not need a subscript $i$. I also wonder whether it would make sense to neglect the shear stress component from the start, as it is found later in section 3.1 to be negligible compared to the tensile and compressive stresses. In this case, the bonds could be modelled as thin beams.

4. In section 3.1, I think it would make more sense to discuss figure 6 before figures 4 and 5, as it provides examples of stress profiles through the ice floe, which helps understand the parametric dependence analysis conducted in the other two figures.

5. I have trouble understanding the decay in vertical displacement and tensile stress with distance from the floe edge observed in the bottom panels of figure 3, in figure 6(a) and in figure (8). The model does not account for dissipative processes and multiple scattering (when multiple floes are present), so I wonder if this decay arises from the numerical scheme, in which case its impact on the results should be discussed, or non-linearities in the wave model, although it would be unlikely considering wave steepness is very small in all cases. The decaying behaviour is mentioned in page 10 (line 30), but its cause is not discussed. In any case, a discussion of this phenomenon is needed.

6. In the second paragraph of section 3.2, the author discusses how neglecting drift and surge motion, and the resulting floe-floe collisions, may lead to underestimating wave attenuation rates and overestimating the extent of the zone of broken ice. Although I agree with this statement, I think it is also important to discuss the effect of neglecting multiple scattering by an array of floes after breaking has occured which, when accounted for, may lead to constructive or destructive interference and therefore affect the attenuation rate. This particular phenomenon is the

focus of the manuscript *Modelling wave-induced sea ice breakup in the marginal ice zone* by Montiel and Squire currently under review (article can be accessed on arXiv at `https://arxiv.org/abs/1705.05941`). The influence of multiple scattering on the breaking pattern is hard to estimate and I do not suggest that author attempts to do it, but I think it should be mentioned as a limitation of the present study.

**Specific comments**

1. page 1, line 1: I suggest to rephrase "the variability of wave-induced stress and breaking in sea ice" by "wave-induced stress and breaking in sea ice for a range of wave and ice conditions".

2. page 1, line 3: I do not think quotes are appropriate for joints or in all subsequent instances throughout the manuscript.

3. page 1, lines 4 and 5: I think "part" should be replaced by "module" for consistency.

4. page 1, line 14: I do not think "defining characteristic" is correct as other processes are important in forming the MIZ and several definitions exist in the literature Consider rephrasing slightly.

5. page 1, line 15: I suggest to replace "ice cover" by "ice-covered ocean" or "sea ice cover" to be more precise.

6. page 1, lines 15–20: these statements relate to the Arctic Ocean only, as it is quite a different story in the Southern Ocean. Make sure to specify this.

7. page 1, line 23: I suggest to replace "show" by "suggest".

8. page 1, line 25: "continuum models" should be defined.

9. page 2, line 2: if the purpose is to have an exhaustive list of studies consider-
ing parametrisations of wave-ice interactions in large scale models, the authors
should also reference Bennetts et al. (2017, The Cryosphere) as well as hind-
casts studies conducted with the spectral wave model WAVEWATCH 3, partic-
ularly Collins et al. (2015, Geophysical Research Letters), Li et al. (2015, Geo-
physical Research Letters), Ardhuin et al. (2016, Geophysical Research Letters)
and Rogers et al. (2016, Journal of Geophysical Research).

10. page 2, lines 5 and 6: rephrase "Due to low temporal ... sea ice conditions". A
brief statement about recent advances in remote sensing techniques to monitor
waves in the MIZ should also be included, as it is currently a very active area of
research.

11. page 2, line 8: I do not think statements like "seemingly basic processes" are ap-
propriate, as there is nothing basic about processes governing the interactions of
ocean waves with sea ice. In addition, the example given on the attenuation rate
of wave height is not a process, but merely an effect of the processes governing
the propagation of waves in the MIZ.

12. page 2, line 11: I do not agree with the list of references given here to support
the argument. Squire et al. (2009) and Vaughan et al. (2009) do not model wave
attenuation in the MIZ but in non-fragmented pack ice, while Dumont et al. (2011)
parametrise wave attenuation using the model by Kohout et al. (2008, Journal
of Geophysical Research). In addition to the latter study, I suggest the following
references: Bennetts and Squire (2012, Proccedings of the Royal Society A),
Mosig et al. (2015, Journal of Geophysical Research) and Montiel et al. (2016,
Journal of Fluid Mechanics), as different approaches to model wave attenuation
in the MIZ.

[Figure]

13. page 2, lines 12 and 13: replace "a storm event" by "a field experiment" and "in height" by "in significant wave height".

14. page 2, line 14: the observations of those breakup events were reported in Kohout et al. (2016, Deep Sea Research Part II).

15. page 2, lines 14–16: This sentence does not belong here. The discussion is about wave attenuation in the MIZ, while this is a general impact statement for wave-ice interactions.

16. page 2, lines 26 and 27: I think it would be appropriate to give a brief description of the "basic mechanisms of wave-induced ice breaking" mentioned here.

17. page 2, line 33 and 34: is the relationship between wavelength and floe size assumed or is it a consequence of the model as in Williams et al. (2013)? Please clarify this statement.

18. page 3, lines 18 and 19: it is unclear what the author means by "prohibiting inelastic effects from becoming significant" Please clarify this statement.

19. page 3, line 27: replace "module" by "modules".

20. page 3, line 30: I think the author should say something about $\sigma$-coordinates or say that they will define them later, as most readers will likely be unfamiliar with them.

21. page 3, line 31: replace "(moving) ice" by "floating ice".

22. figure 1: I find some aspects of the figure to be slightly misleading. More specifically, pressure and velocity points in the ice grains suggest a vertical variation of these quantities through the ice thickness, which is obviously not the case. I suggest that the author removes those and denote the center of mass instead.

Further, the situation in which an area of open water exists between two ice floes, as shown in the figure, is not considered in any of the simulations conducted in this study, so I think it would be sensible not to show this situation in the figure. I also think that it would be useful to include a sketch of the bonds in a bent situation, either on Figure 1 or in a separate figure. It would help understand their description at the beginning of page 5.

23. page 4, lines 29–31: the first part of this sentence is oddly constructed and hard to read. Also I do not agree with the conclusion reached in the second part of the sentence, as drift and surge motion do not depend on the compactness of the MIZ (or concentration). The significance of these phenomena depends on floe length and thickness, wavelength and incident wave amplitude.

24. page 5, equations (2)–(4): even though most readers will likely recognise these equations, I think it would be useful to introduce them, i.e. say what they mean.

25. page 6, equation (5): replace $h$ by $H$.

26. page 6, below equations (11)–(17): I think that the author should introduce briefly the hydrodynamic forcing terms here, even though they are fully described later in section 2.2.3.

27. page 7, lines 4 and 5: I do not understand the meaning of this sentence. In any case, the statement seems to apply only when grain width varies from grain to grain, which is not the case here, so I wonder whether this statement is necessary.

28. page 7, lines 18 and 19: can the author give a reference to support this statement?

29. page 8, equations (8) and (9): these equations only account for the fluid pressure on the bottom surface of each grain. Does the author use modified formulae for

the end grains, which have a side surface in contact with the fluid? The contribution of the pressure field acting on these surfaces will modify the force and moment on the end grains.

30. page 8, equations (27) and (28): the parameter $d$ in these equations is not defined.

31. Table 1: can the author explain the choice of tensile strength, as it approximately one order of magnitude smaller than typical values for sea ice (see, e.g., Timco and Weeks, 2010, particularly Figure 7 therein)?

32. page 9, line 15: does the author mean "modelled ice floe" instead of "model ice"?

33. page 9, lines 20 and 21: I think the author should introduce what is plotted in figure 3 in the text. Also please discuss the attenuation of $z_i$ and $\sigma_t$, as suggested in my earlier comment.

34. page 9, lines 24 and 25: this statement is a bit too simplistic and not representative of what is seen in Figure 4(c). The figure shows $\sigma_{t,max}$ plateaus beyond a critical floe size, but the latter depends on thickness in a non-trivial way, as it is about 20, 100 and 50 m for $h = 0.5$, 1 and 2 m, respectively, so I am a bit confused by the statement that it is "equal to approximately two wavelengths", which is about 80 m.

35. page 9, lines 27 and 28: In addition to the large reflection, the author should mention that thicker floes tend to behave more like rigid bodies with lower strains (or curvature of the deflection function) and therefore decreased stresses.

36. page 9, lines 28 and 29: I think the author means that the location of $\sigma_{t,max}$ is independent on thickness, not $\sigma_{t,max}$ itself. It would also be useful to have seen Figure 6 before to understand this statement better, as suggested in my comment earlier.

37. page 10, lines 3–6: Again, I think this explanation could be improved by saying that at low and high frequencies, rigid body motions dominate over flexural motions, as demonstrated for instance by Montiel et al. (2013, Journal of Fluid Mechanics).

38. page 10, line 10: please rephrase "In the floe interior", e.g. "Sufficiently far from the ice edge" or something similar.

39. page 10, line 11: can the author explain these ripples? Is it numerical or physical?

40. page 10, line 12: This statement seems to hold when the floe length is less than half the wavelength. Maybe the author could mention that.

41. page 10, line 17: I don't understand the statement "An individual wave is responsible for a few breaking events". Could the author clarify?

42. page 10, line 18: what is meant by "weaker ice"? Is it small thickness, tensile strength, ...?

43. page 11, line 1: can the author define or at least briefly describe a "Jonswap energy spectrum"?

44. page 11, line 15: I do not agree that breaking does not depend on "the characteristics of the incoming waves", as clearly it will depend on wave amplitude. We can probably expect a linear relationship between incident wave amplitude and stress. This comment also applies to a similar statement in line 27.

45. page 11, lines 17 and 18: main result (iii) is rather obvious as there is not much else the stress can depend on!

46. page 11, lines 19 and 20: the author has not introduced the concept of "breaking probability" earlier in the manuscript, so it is confusing to have it in the conclusion. Maybe this could be slightly rephrased.

47. page 11, line 22: replace "might be not realistic" by "might not be realistic".

48. page 11, lines 27 and 28: I think the author should also mention multiple scattering here, as discussed in my earlier comment.

49. page 11, lines 27–29: can the author include a reference to support the statement in brackets?

50. page 11, lines 29–32: I am having trouble agreeing with this statement, as I think the incident wave amplitude plays a determining role in creating the floe size distribution, i.e. the larger the wave amplitude the smaller the floes.

---

## Referee Comment (RC2) · Anonymous Referee #2 · 23 Jun 2017

**1  General comments**

This is generally a good paper, with some interesting conclusions. In addition, the tool developed by the author has a lot of potential for further investigations when all the effects available in the Design model (collisions) and NHwave (eg. turbulence) are included in this coupled code.

[Figure]

**2 Specific comments**

1. p2: *'For example, the functional form describing the rate of change of wave height with distance from the ice edge is far from established. . . '* Meylan et al (2014) found exponential decay from this dataset when the individual frequencies are considered instead of the significant wave height. Li et al (2015) were able to qualitatively produce the linear decay of the significant wave height with Wavewatch 3 — proposing that the nonlinear source term $S_{nl}$ was the reason, as it moves energy to lower frequencies which are attenuated more slowly.

2. p8: please give the equations for the free-slip boundary conditions.

3. equation (7): Ma et al (2012) has the total derivative of $w$ in the bottom condition $(\mathrm{d}w/\mathrm{d}t)$. Maybe this is a typo? Similarly, perhaps the pressure condition at the lower surface of the ice (21) should be checked?

4. p6: How are the waves generated?

5. *§3.1 Stress variability in continuous ice*

    (i) The locations of $\sigma_{t,max}$ seem to occur about 10 m away from the ice edge, which seems very small — eg. Squire et al (1995): *'Anecdotally it appears that incoming waves and swell cause a fracture line to develop a few tens of meters back from the ice margin and parallel to it'*. Was there any attempt to tune the wave number in the ice to a realistic value, eg. for a thin elastic plate? Perhaps these numbers would increase if such tuning were done.

    (ii) Presumably the damping of the waves is due to some damping in the bonds, although this is not mentioned anywhere. Perhaps if this was removed it could be useful to see how much of the attenuation is numerical and how much is physical. The reflection and transmission coefficients could also be compared to linear models for a semi-infinite elastic plate. It would also be interesting to see

if this affects the conclusions about the location of $\sigma_{t,max}$ being independant of wavelength or not.

6. *§3.2 Breaking of uniform ice by regular waves*

    (i) It would be interesting to see fig 8 without damping in the bonds and also the time evolution of the wave amplitude to see if there is more or less attenuation after breaking — on one hand the floes produced are small compared to the wavelength but on the other hand there are many of them and perhaps multiple scattering could do something.

**References**

Jingkai Li, Alison L. Kohout, and Hayley H. Shen. Comparison of wave propagation through ice covers in calm and storm conditions. *Geophysical Research Letters*, 42(14):5935–5941, 2015. 2015GL064715.

Gangfeng Ma, Fengyan Shi, and James T Kirby. Shock-capturing non-hydrostatic model for fully dispersive surface wave processes. *Ocean Modelling*, 43:22–35, 2012.

M. H. Meylan, L. G. Bennetts, and A. L. Kohout. In-situ measurements and analysis of ocean waves in the Antarctic marginal ice zone. *Geophys. Res. Lett.*, 41(14):5046–5051, 2014.

V. A. Squire, J. P. Dugan, P. Wadhams, P. J. Rottier, and A. J. Liu. Of ocean waves and sea ice. *Annu. Rev. Fluid Mech.*, 27:115–168, 1995.

---

## Referee Comment (RC3) · A. Kulchitsky (Referee) · 14 Jul 2017

**1   General Comments**

The paper describes 2D discrete element method (DEM) model of sea ice coupled with existing NHWAVE code for solving incompressible dynamics Navier-Stocks equations for water. The DEM uses rectangular grid from NHWAVE discretization to represent ice floes as a set of rectangular cells bonded together. Bonds are represented by a virtual rectangular as well with mechanical properties similar to elastic beams.

The NHWAVE and DEM parts are coupled using the boundary conditions. NHWAVE

uses the ice velocity at the boundary with ice elements using no-slip condition. DEM model uses force and torque caluclated from water pressure at the ice surface.

Closed system of equations is solved with each part using different time steps as characteristic time of DEM model is 2 orders of magnitude smaller. The results show how a single ice floe interact with waves, in particular where the maximum stress appears in the floe and how the floe is breaking under the waves.

The paper is well written although I had some significant problems with all the notations introduced that either are not explained, explained too remote from the equations, or not shown.

I support publishing this manuscript after some corrections. As I am a DEM specialist, I can mostly adress only the DEM part of the publications in my comments.

**2 Specific Comments**

**2.1 Substantial issues**

1. My main concern about this work is the lack of verfication of the approach and the code. For example, does discretization step $\Delta x$ change the outcome of the computations? It is unclear until some scaling tests are done. I would suggest to use different $\Delta x$ with the same wave input and compare the results. *This can be corrected within this paper by running some tests with $\Delta x$ being a half of the $\Delta x$ used in the paper and comparing the results. Maybe a reference that the final results do not change with $\Delta x$ within the numerical accuracy would be sufficient.*

2. I have some concerns regarding 2D approach to the problem of breaking an ice floe. It is just hard to imagine how to interpret the results of a single ice flow breaking in long ($y$-dimention) but narrow ($x$-dimension) fragments as the results

of simulatons show. Actual floe would break along $x$ dimension as well such that it will be easier to break later. I suppose 2D approach is still applicable for testing in channels. *This issue is not to be corrected and just brougt to the discussion at this point.*

**2.2 More specific comments**

1. I struggled to understand the 2.1 and 2.2 sections with just Figure 1. It would be very useful to add dimensions, indices and more notes to the figure. In this case the reader would see what $\Delta x$, $l_i$, etc. are. Maybe a single cell scheme with forces and torques shown, as well as velocities, constraints ($u_i = 0$) and bonds.

2. Sec. 2.1, p. 5, 0–5. "All bonds are cuboid". Should it say "All bonds are elastic"?

3. Equations (13) and (14) include terms for the torques and forces imposed from water ($M_{wv,i}$ and $F_{wv,i}$) that are only explained 2 pages later in the coupling section. They should be defined after the equations as well.

4. I have some questions regarding "classical beam theory" equations (18)–(20). First, the length of the bonds is defined generally as $l_{b,i} = \lambda\Delta x$ (page 5, 0–5). Please, verify that $l_{b,i} = h_{b,i}$ for these computations. Moreover, I suggest to introduce a picture showing all the stresses on the bonds.

---

## Editor Comment (EC1) · J. Hutchings (Editor) · 14 Jul 2017

Dear Dr. Herman,

All of the expected reviews are now in for this paper. It is possible more comments may be received while the paper is in open discussion. Once the open discussion is closed you will received instructions, and in the mean time I recommend you respond to the reviewers, with a manuscript revision in mind.

Best regards, Jenny
* * *

---

## Author Comment (AC1) · 28 Aug 2017

**General comments**

First of all, I would like to thank the Reviewers for very insightful comments and valuable suggestions for corrections and modifications to the manuscript. I would also like to thank the Editors for selecting reviewers whose expertise covers both theoretical aspects of the manuscript, those related to the physics of wave-ice interactions, as well as numerics.

Below, I first reply to general issues raised by at least two Reviewers, and I also describe the major changes that have been made to the manuscript. In the further parts of this document, I reply to the particular comments of individual Reviewers.

In the revised manuscript, modified/new text is typeset in blue.

1. The most important general issue raised by the Reviewers seems to be the simulated wave attenuation clearly seen in the results and not discussed in sufficient detail in the text. The wave model was run without any dissipation mechanism (although it is worth mentioning that various such mechanisms, e.g., bottom friction or a turbulence model, are implemented in NHWAVE and can be readily used in the coupled model if necessary). NHWAVE often produces artificial attenuation, for example if the horizontal resolution of the model is too coarse relative to the modelled wavelengths, or when the number of layers is too small to reproduce the vertical structure of the wave motion. However, the model configuration used in the simulations presented in this manuscript was tested by running the wave model without ice – and no such effects were present. Therefore, as the Reviewers correctly remarked, the attenuation visible in the results takes place in the ice. (Another possible source of damping is in the coupling mechanism, e.g., when the coupling time step is not small enough — however, this was not the case in the present simulations.)

The numerical scheme used to solve the sea ice equations leads to damping, so that, effectively, the bonds do not behave perfectly elastic. The intensity of damping depends on wavelength – shorter waves get damped more than long waves – as well as the time step used. These are clear drawbacks and I'm going to improve the present scheme or replace it with a more robust one in the future versions of the model. Interestingly, it is a standard scheme used in the LIGGGHTS discrete-element model, and thus also used in the DESIgn sea ice toolbox. Apparently, the damping manifests itself particularly strongly in simulations with cyclic, fast time-varying forcing – as is the case in simulations with ice moving on waves. This is the reason why the problem remained unnoticed in previous applications of the DESIgn model done by myself and others. (Even more than that: in the default code of LIGGGHTS, an additional damping coef-
ficient is used, as it is assumed that it reduces spurious oscillations in bonds, without negatively affecting modelling results!)

However, it must be stressed that the wave attenuation within the ice plays a very minor role from the point of view of the analysis presented in this paper. Ice breaking takes place in a narrow zone close to the edge of the unbroken ice, where the effects related to the presence of this edge lead to enhanced wave amplitude and therefore strain. I performed tests in which the damping within bonds was artificially modified (by introducing a damping coefficient to the bond equations (15)–(17) and varying its value) and in terms of the floe sizes the results were exactly the same. In other words, reducing damping in the bonds is undoubtedly crucial for future applications of the model, but it does not affect the results discussed here. Nevertheless, I agree with the Reviewers that this issue should be clearly stated in the paper. I added information about damping to Section 3.1, in which issues related to the model setup are discussed, as well as to the discussion section at the end of the paper – please see the text of the revised manuscript.

The Reviewers also asked about the choice of the model parameters (ice strength etc.). As I said in the paper, I did not attempt to calibrate the values of the parameters to any real-world situation. Apart from the results that are presented in the manuscript, I made several more simulations with widely varying ice and wave characteristics, and none of these results affected the conclusions of the manuscript. The set of model parameters described in the paper is simply the one for which I collected the largest, the most complete set of results. This choice was affected by my present work, in which the model is used to reproduce the results of a laboratory experiment on ice breakup by waves. Thus, the fact that the wave amplitude in the simulations presented in the paper is so small and the ice strength so low can be seen as an "artefact" of this work. But once again: the results are very robust, and even though different model parameters would certainly produce different numbers (e.g., larger floes, with lengths closer to those reported from the field), the conclusions would remain the same.
**2.** In response to the critics of the Reviewers, I rewrote/rearranged large parts of the Introduction – even though the modifications do not always follow the detailed comments of the Reviewers. In general, I tried to follow the advice to make the introduction more "fit to purpose". In particular, I removed the text on wave attenuation in the MIZ – first, because many parts of it were (rightfully!) criticized by both Reviewer 1 and 2, and second, because it is only broadly relevant to the topic of this paper. Also, the first part of section 2, providing a general introduction to the two models, has been moved to section 1, according to the suggestion of Reviewer 1. See the revised manuscript for the new version of the text, and my replies to individual comments below.

**3.** I added a new figure, showing top and side views of two grains connected with a bond, in order to illustrate the model components and definitions of the model properties in Section 2.1.

**REPLY TO THE COMMENTS OF REVIEWER No. 1 (F. Montiel)**

My main concern is that the article lacks a detailed discussion on the advantages and drawbacks of this new modelling approach compared to existing waves/ice interactions models. For instance, it would be good to have an idea of the computational complexity of the model compared to others and the main parametric restrictions affecting this complexity, e.g. small water depth. I think it would have been nice to conduct a comparative study of wave propagation through an ice floe with an existing model for validation, but I understand this is not what this paper is about.

Yes, this is beyond the scope of this paper – see point 1. above.

I don't understand the comment on the "small water depth". The water depth doesn't have to be small! The NHWAVE model does not make any assumptions about water depth. Moreover, instead of constant thickness of  $\sigma$ -layers, the water column can be divided into  $\sigma$ -layers logarithmically, so that if the water depth is large, the layers can be very thin close to the surface and thicker towards the bottom, reflecting the structure

TCD
of the deep-water wave.

NHWAVE is a very flexible, nonlinear hydrodynamic model – and all its advantages can be easily used in coupled wave–ice simulations. In particular, there is no assumption regarding irrotational motion (as many wave–ice interaction models assume) or the form of the wave forcing – it does not have to be a sum of sine waves, any nonperiodic time series can be used as well. Additionally, NHWAVE can handle different types of boundary conditions (vertical, at the bottom etc.).

If the sea ice module is concerned, the manuscript discusses the limitations of the present version (e.g., the lack of motion in the horizontal plane; Section 2) and the plans of the future development (Section 4). In the revised manuscript, I extended the last paragraph of Section 4 so that it provides more details relevant to your questions. I also added some comments on (extremely high) computational costs of this modelling approach.

**General comments**

1. I think the Introduction section needs some work. To be more specific, I think it could be re-organised more efficiently so it is more relevant to the scientific question addressed in the rest of the manuscript. The first paragraph discusses (i) the geophysical context, i.e. Arctic sea ice reduction and the impact of ocean waves, (ii) the parametrisation of wave/ice interactions in large scale forecasting models, (iii) the limited observational data, (iv) the attenuation rate of wave height in the MIZ and (v) the impact of waves on sea ice in the Southern Ocean. These are all relevant discussions, but they each need to be expanded and separated in different paragraphs. The second paragraph, on the other hand, lists the range of topics and associated publications on wave/ice modelling in the literature, even though it is not clear how they are relevant (if at all) to the present manuscript. I would suggest a more fit-to-purpose approach to the Introduction, in which the authors state their goals early on, and then discuss what has already been done to address these questions and how the new model proposed
here goes beyond what is already established. I also suggest that the two paragraphs at the beginning of section 2, and before the start of section 2.1, are integrated as part of the Introduction, as they give a broad presentation of the two models used in this study, which is lacking in the current version of the Introduction. Additional comments on specific parts of the Introduction are given below.

Regarding the first part of this comment: The first paragraph of section 1 contains a general introduction (with topics listed by the Reviewer); the second and third paragraphs describe the literature related to SEA ICE BREAKING, not to sea ice–wave interactions in general. Yes, problems other than ice breaking are mentioned, but only two sentences are used for that (I admit, they are rather long). In my opinion, it's the first part of the introduction that can be argued to be not very relevant to the content of this paper, not paragraphs two and three. In particular, the quotes from Squire (1995) are crucial for the results presented in this paper – and they are part of the second paragraph, criticized by the Reviewer, who suggests expanding the first paragraph and reducing the following ones. Whereas I fully agree that the introduction should be made more "fit to purpose", my understanding of how this should be done seems to be opposite to that suggested by the Reviewer – see my general comment No. 2 above.

I followed the suggestion to move the first part of Section 2 to Section 1. I combined it with the text of the next to the last paragraph.

If making the introduction more "fit to purpose" is concerned, I added information on how ice breaking is treated in the existing parameterizations used in large-scale models, so that it is clear that breaking in these models is parameterized, not simulated explicitly. Please see the new introduction for the changes/additions that have been made.

2. As mentioned earlier, the manuscript needs to discuss in more depth other modelling studies on wave-induced breakup in the MIZ, particularly the papers referenced in page 2 (line 2), to which should be added the recent study by Bennetts et al. (2017) in The
Cryosphere, in which the breakup is treated in a very different manner to that of the present model. More focus should then be added on how the present model differs from these existing models and what are its relative strengths and weaknesses.

But the models in question parameterize the effects of breaking, they do not simulate breaking itself! I state this very clearly both in the new introduction and in the discussion section. In fact, the recent paper by the Reviewer (Montiel and Squire 2017) is the only one in which ice breaking by waves is in the focus of the analysis and in which no a priori assumptions on the FSD resulting from breaking is made! I added a reference to that paper to the revized introduction.

3. The model described in section 2 is presented with much generality, and by doing so, the author introduced many variables that are later either neglected or set to be constant. I understand that the author wants to convey the generality of the modelling approach described here, but considering the main goals of the present study is to analyse the tensile stress in the bonds and breaking of a single floe with uniform properties, I do not think it is necessary to overcomplicate the equations. More specifically, the turbulent diffusion terms in equations (3) and (4) are not needed and the quantities associated with the bonds and grains that are constant, e.g. thickness, width, mass, moment of inertia and elasticity parameters, do not need a subscript i. I also wonder whether it would make sense to neglect the shear stress component from the start, as it is found later in section 3.1 to be negligible compared to the tensile and compressive stresses. In this case, the bonds could be modelled as thin beams.

I understand these objections, and I myself spent some time considering the advanteges and disadvantages of these two alternatives – adjusting the model description to the configuration used in the paper, or keeping it more general in order to illustrate a wider range of applications of the model. I decided to keep the model description more general. I don't think that introducing grain and bond indices in the sea ice equations introduces significant complications to the model description or make it less clear – especially that many grain and bond variables (instantaneous forces, stresses, velocities,
etc.) need an index anyway, as they vary from grain to grain and from bond to bond.

As for the shear stress in the bonds: I don't see a good reason for making any a priori assumptions regarding its amplitude. I think it is more valuable to have the relative "importance" of different stress components as a result of the simulations rather than to start with simply disregarding some components. If the shear stress was simply set to zero, one could ask how do I know that it does not contribute to the breaking process...

4. In section 3.1, I think it would make more sense to discuss figure 6 before figures 4 and 5, as it provides examples of stress profiles through the ice floe, which helps understand the parametric dependence analysis conducted in the other two figures.

Figures 4 and 5 (or, according to the new numbering in the revized manuscript, 5 and 6) are related to the first, major stress maximum close to the ice edge. I think that the knowledge of the stress variability along the whole floe is not necessary to the understanding of how the value and location of this major stress maximum changes with ice thickness or incomming wavelength. I find the version you propose, with a description of the stress variability across ice floes followed by the analysis of the stress maximum, equally logical and justified. I decided to keep the first version, only because I don't see any important reasons to change it.

However, I made a number of small modifications to this part of the text, following your detailed comments (see further).

5. I have trouble understanding the decay in vertical displacement and tensile stress with distance from the floe edge observed in the bottom panels of figure 3, in figure 6(a) and in figure (8). The model does not account for dissipative processes and multiple scattering (when multiple floes are present), so I wonder if this decay arises from the numerical scheme, in which case its impact on the results should be discussed, or non-linearities in the wave model, although it would be unlikely considering wave steepness is very small in all cases. The decaying behaviour is mentioned in page 10 (line 30), but
its cause is not discussed. In any case, a discussion of this phenomenon is needed.

See my general comment on attenuation in the first part of this document.

6. In the second paragraph of section 3.2, the author discusses how neglecting drift and surge motion, and the resulting floe-floe collisions, may lead to undersestimating wave attenuation rates and overestimating the extent of the zone of broken ice. Although I agree with this statement, I think it is also important to discuss the effect of neglecting multiple scattering by an array of floes after breaking has occured which, when accounted for, may lead to constructive or destructive interference and therefore affect the attenuation rate. This particular phenomenon is the focus of the manuscript Modelling wave-induced sea ice breakup in the marginal ice zone by Montiel and Squire currently under review (article can be accessed on arXiv at https://arxiv.org/abs/1705.05941). The influence of multiple scattering on the breaking pattern is hard to estimate and I do not suggest that author attempts to do it, but I think it should be mentioned as a limitation of the present study.

Yes, I agree that the effects of scattering might be important and are hard to estimate.

I added a comment on multiple scattering to the text in the second paragraph of section 3.3 (previously 3.2). I also made a comment that the model is capable of simulating multiple scattering, but not in the configuration that was used in this paper – see the modified manuscript.

**Specific comments**

1. page 1, line 1: I suggest to rephrase "the variability of wave-induced stress and breaking in sea ice" by "wave-induced stress and breaking in sea ice for a range of wave and ice conditions".

Changed as suggested.

2. page 1, line 3: I do not think quotes are appropriate for joints or in all subsequent instances throughout the manuscript.
That's true, the quotes have been removed.

3. page 1, lines 4 and 5: I think "part" should be replaced by "module" for consistency.

Changed as suggested.

4. page 1, line 14: I do not think "defining characteristic" is correct as other processes are important in forming the MIZ and several definitions exist in the literature. Consider rephrasing slightly.

The sentence says it is "A defining characteristic", not "THE difining characteristic". It doesn't suggest that it is the only one. Although the term MIZ is used in a different way by different authors (as the second part of the sentence in question says), I don't think anyone would claim that the influence of waves is not an inherent feature of MIZ.

5. page 1, line 15: I suggest to replace "ice cover" by "ice-covered ocean" or "sea ice cover" to be more precise.

Changed as suggested.

6. page 1, lines 15-20: these statements relate to the Arctic Ocean only, as it is quite a different story in the Southern Ocean. Make sure to specify this.

The phrase "in polar and subpolar regions" has been extended to "in polar and subpolar regions of the northern hemisphere".

7. page 1, line 23: I suggest to replace "show" by "suggest".

Changed as suggested.

8. page 1, line 25: "continuum models" should be defined.

I added the text "(i.e., those in which ice is treated as a continuous mass rather than as discrete particles)" to this sentence.

9. page 2, line 2: if the purpose is to have an exhaustive list of studies considering parametrisations of wave-ice interactions in large scale models, the authors should
also reference Bennetts et al. (2017, The Cryosphere) as well as hindcasts studies conducted with the spectral wave model WAVEWATCH 3, particularly Collins et al. (2015, Geophysical Research Letters), Li et al. (2015, Geophysical Research Letters), Ardhuin et al. (2016, Geophysical Research Letters) and Rogers et al. (2016, Journal of Geophysical Research).

Do you mean the paper by Collins et al. titled "In situ measurements of an energetic wave event in the Arctic marginal ice zone"? This one is a case study in which a SWAN model is used that does not take into account any ice-related effects. No parameterizations of wave-ice interactions are proposed. The papers of Li et al. (2015), Ardhuin et al. (2016) and Rogers et al. (2016) present very interesting and valuable results on wave propagation in sea ice, but they use the existing parameterizations that are implemented in the Wavewatch model – and this is what I should have cited, I think. I added a reference to Tolman and the WAVEWATCH III® Development Group (2014) to the revised text, as well as a reference to Bennetts et al. (2017).

10. page 2, lines 5 and 6: rephrase "Due to low temporal ... sea ice conditions". A brief statement about recent advances in remote sensing techniques to monitor waves in the MIZ should also be included, as it is currently a very active area of research.

This sentence is a statement about the TEMPORAL RESOLUTION of remote sensing data from polar regions. I'm not an expert in remote sensing, but to the best of my knowledge, temporal resolution of satellite data at high latitudes remains a problem. In their recent review paper by Ardhuin et al. (2017, Ocean Science Discuss.), they write that "Ice concentration is the only parameter that is well monitored near the ice edge" (section 4.2). I slightly modified the next sentence so that the advances in remote sensing methods are briefly mentioned (see text).

11. page 2, line 8: I do not think statements like "seemingly basic processes" are appropriate, as there is nothing basic about processes governing the interactions of ocean waves with sea ice. In addition, the example given on the attenuation rate of
wave height is not a process, but merely an effect of the processes governing the propagation of waves in the MIZ.

I agree that the statement is unfortunate. I rewrote it. I also removed the text following it, as already mentioned in my comments above.

12. page 2, line 11: I do not agree with the list of references given here to support the argument. Squire et al. (2009) and Vaughan et al. (2009) do not model wave attenuation in the MIZ but in non-fragmented pack ice, while Dumont et al. (2011) parametrise wave attenuation using the model by Kohout et al. (2008, Journal of Geophysical Research). In addition to the latter study, I suggest the following references: Bennetts and Squire (2012, Proccedings of the Royal Society A), Mosig et al. (2015, Journal of Geophysical Research) and Montiel et al. (2016, Journal of Fluid Mechanics), as different approaches to model wave attenuation in the MIZ.

This text (the whole paragraph) has been removed (see my general comments above).

13. page 2, lines 12 and 13: replace "a storm event" by "a field experiment" and "in height" by "in significant wave height".

This text has been removed.

14. page 2, line 14: the observations of those breakup events were reported in Kohout et al. (2016, Deep Sea Research Part II).

This text has been removed.

15. page 2, lines 14-16: This sentence does not belong here. The discussion is about wave attenuation in the MIZ, while this is a general impact statement for wave-ice interactions.

This text has been removed.

16. page 2, lines 26 and 27: I think it would be appropriate to give a brief description of the "basic mechanisms of wave-induced ice breaking" mentioned here.

TCD
I extended the sentence to: "basic mechanisms of wave-induced ice breaking, related to the presence of secondary ice-coupled waves affecting the wave envelope close to the ice edge and rapidly decaying away from the edge."

17. page 2, line 33 and 34: is the relationship between wavelength and floe size assumed or is it a consequence of the model as in Williams et al. (2013)? Please clarify this statement.

In Williams et al. (2013) the relationship is not a "consequence of the model", as you suggest, but it is assumed! At the beginning of their section 3.2.2 they write: "If it [the ice] breaks, the maximum floe size is set to  $D_{max} = \max(\lambda_w/2, D_{min})$ ". The statement is correct.

18. page 3, lines 18 and 19: it is unclear what the author means by "prohibiting inelastic effects from becoming significant" Please clarify this statement.

All elastic plate models of sea ice are based on this assumption. My statement is almost a direct quote from Fox and Squire's paper ("elasticity is justified because of the oscillatory nature of the problem which does not allow anelastic processes to act in any significant way.").

19. page 3, line 27: replace "module" by "modules".

The phrase has been changed to "the sea ice module and the wave module".

20. page 3, line 30: I think the author should say something about  $\sigma$ -coordinates or say that they will define them later, as most readers will likely be unfamiliar with them.

I added "(see further Section 2.2.1)" at the end of this sentence.

21. page 3, line 31: replace "(moving) ice" by "floating ice".

Changed as suggested.

22. figure 1: I find some aspects of the figure to be slightly misleading. More specifi-

TCD
cally, pressure and velocity points in the ice grains suggest a vertical variation of these quantities through the ice thickness, which is obviously not the case. I suggest that the author removes those and denote the center of mass instead. Further, the situation in which an area of open water exists between two ice floes, as shown in the figure, is not considered in any of the simulations conducted in this study, so I think it would be sensible not to show this situation in the figure. I also think that it would be useful to include a sketch of the bonds in a bent situation, either on Figure 1 or in a separate figure. It would help understand their description at the beginning of page 5.

As for the first part of this comment: the wave module DOES solve the hydrodynamic equations within the ice as well. This is the core idea – and the strength – of immersed boundary methods, especially in applications in which the submerged objects are moving relative to the grid of the hydrodynamic model (so that "wet" and "dry" grid cells change from time step to time step). Solving the Poisson equation for pressure is computationally expensive, and therefore it is beneficial if the coeffcient matrix does not have to be rebuilt at each time step, but can be built only once instead. In immersed boundary methods, the influence of boundaries is handled through the source terms – see the papers I'm citing for details – and the values of pressure are determined in each grid point, independently of it being "wet" or "dry" at the given time instance. But of course, the points inside submerged bodies are not considered when the modeling results are analyzed. I added a short information on that to the caption of Fig. 1.

As for the remaining parts of this comment: As I wrote in the first, general part of my reply, I decided to keep the description of the model (and consequently also the Figure illustrating it) general. I also added a figure illustrating two grains connected with a bond, with relevant variables etc.

23. page 4, lines 29-31: the first part of this sentence is oddly constructed and hard to read. Also I do not agree with the conclusion reached in the second part of the sentence, as drift and surge motion do not depend on the compactness of the MIZ (or concentration). The significance of these phenomena depends on floe length and
thickness, wavelength and incident wave amplitude.

As I understand it, at 100% ice concentration the ice cover may drift as a whole, but the oscillatory surge motion (i.e., different horizontal velocity of neighboring ice floes) would require ridging or rafting to occur in areas of convergence. Last year I took part in a laboratory experiment of (broken) ice motion on waves, and we recorded almost perfectly vertical motion of the ice, with vanishing horizontal velocity component – as long as the ice was compact.

I rephrased slightly the first part of this sentence, so that it is easier to read.

24. page 5, equations (2)-(4): even though most readers will likely recognise these equations, I think it would be useful to introduce them, i.e. say what they mean.

I added a short comment that "the governing equations are the mass and momentum conservation equations".

25. page 6, equation (5): replace h by H.

Corrected.

26. page 6, below equations (11)-(17): I think that the author should introduce briefly the hydrodynamic forcing terms here, even though they are fully described later in section 2.2.3.

I moved the sentences introducing these terms ("Finally, the first terms on the righthand-side of (13) and (14) denote the net moment of forces and the net vertical force, respectively, from the wave motion underneath the ice. They are calculated by integrating the contribution from waves over the wetted surface of the grains. Their detailed formulation is given further in Section 2.2.3.") to the paragraph directly following Eqs. (11)-(17).

27. page 7, lines 4 and 5: I do not understand the meaning of this sentence. In any case, the statement seems to apply only when grain width varies from grain to grain,

TCD
**which is not the case here, so I wonder whether this statement is necessary.**

The torque, being a product of force and position vector, is different for grains of different sizes, even if the force transmitted by the bond connecting those grains is the same. I think this sentence is clear.

**28. page 7, lines 18 and 19: can the author give a reference to support this statement?**

In fact, essentially all "brittle" materials are not prefectly brittle and exhibit a certain level of softening when the stress acting on them exceeds their strength, i.e., the stress does not drop to zero instantaneously, but within a certain time t > 0. As I noted in the revised text, some DEM models take this softening into account in order to obtain more realistic breaking behavior. I cited a review paper by Lisjak and Grasselli (2014).

29. page 8, equations (8) and (9): these equations only account for the fluid pressure on the bottom surface of each grain. Does the author use modified formulae for the end grains, which have a side surface in contact with the fluid? The contribution of the pressure field acting on these surfaces will modify the force and moment on the end grains.

I suppose you mean Eqs. (25) and (26). The contribution of pressure acting on the vertical surfaces of end grains is crucial for the horizontal momentum balance of those grains – which is not solved in this version of the model, as it is assumed that  $u \equiv 0$ . Of course, as the grains move on the water surface, their tilt  $\theta_i$  is different from zero, so that the contribution of pressure acting on vertical grain surfaces to the *vertical* force is not exactly zero. It is proportional to  $\sin \theta_i$  and to the surface area of the vertical wall of the grain, whereas the pressure-related force acting on the lower surface (given by Eq. 28) is proportional to  $\cos \theta_i$  and surface area of that surface, i.e., it is much larger. But of course the contribution from vertical surfaces can be easily added to the formulae (25) and (26).

30. page 8, equations (27) and (28): the parameter d in these equations is not defined.
It's a mistake, thank you for pointing this out! It should be  $\Delta x$  instead. It has been corrected in the revised manuscript.

31. Table 1: can the author explain the choice of tensile strength, as it approximately one order of magnitude smaller than typical values for sea ice (see, e.g., Timco and Weeks, 2010, particularly Figure 7 therein)?

See my general comments at the beginning of this document.

32. page 9, line 15: does the author mean "modelled ice floe" instead of "model ice"?

I think it amounts to the same thing in this context, but I changed it as suggested.

33. page 9, lines 20 and 21: I think the author should introduce what is plotted in figure 3 in the text. Also please discuss the attenuation of  $z_i$  and  $\sigma_t$ , as suggested in my earlier comment.

I modified this fragment so that it includes more details about the content of Fig. 3.

As for attenuation, see my general comment No. 1.

34. page 9, lines 24 and 25: this statement is a bit too simplistic and not representative of what is seen in Figure 4(c). The figure shows  $\sigma_{t,max}$  plateaus beyond a critical floe size, but the latter depends on thickness in a non-trivial way, as it is about 20, 100 and 50m for h = 0.5, 1 and 2 m, respectively, so I am a bit confused by the statement that it is "equal to approximately two wavelengths", which is about 80 m.

I added the text "the stress saturates to a value specific for the given ice thickness", to make it clear that *for a given ice thickness* the value of stress does not change up from a certain floe size. I also changed "two wavelengths" to "one to two wavelengths", as this is where the curves in Fig. 4c,d become approximately horizontal.

35. page 9, lines 27 and 28: In addition to the large reflection, the author should mention that thicker floes tend to behave more like rigid bodies with lower strains (or curvature of the deflection function) and therefore decreased stresses.

TCD
I added this information to the revised text.

36. page 9, lines 28 and 29: I think the author means that the location of  $\sigma_{t,max}$  is independent on thickness, not  $\sigma_{t,max}$  itself. It would also be useful to have seen Figure 6 before to understand this statement better, as suggested in my comment earlier.

Yes, I meant the location of  $\sigma_{t,max}$ . I corrected this.

*37.* page 10, lines 3-6: Again, I think this explanation could be improved by saying that at low and high frequencies, rigid body motions dominate over flexural motions, as demonstrated for instance by Montiel et al. (2013, Journal of Fluid Mechanics).

I added this reference to the text and modified it so that it is clear that both wave reflection and the response of the ice itself are responsible for this effect.

38. page 10, line 10: please rephrase "In the floe interior", e.g. "Sufficiently far from the ice edge" or something similar.

Changed as suggested.

39. page 10, line 11: can the author explain these ripples? Is it numerical or physical?

As I understand it, they are physical, and they are in fact analogous to modulations of the wave envelope at the upwave end of a floe, close to the ice edge (that are observed and predicted by models).

There is a paper by Yoon et al. (2014), in which they analyze experimentally and numerically the motion of floating plates on waves. They observe (and predict with their model) an increase of the amplitude of the vertical motion at the downwave end of the plate (see e.g. their Figs. 7a, 11a, 12b – for the plates without hinges). Figs. 3 and 4 in Kohout et al. (2007) show a similar pattern. As in both papers the model is compared with laboratory experiments, the range of the parameters considered (wavelength, floe size, etc.) is limited by the setup of the experiment. In particular, the ratio of the plate length to the wavelength is relatively small, and the effects of attenuation are much

TCD
weaker, so that there is no smooth region between the fluctuations at the upwave and downwave end of the plates (like in our case with  $L_o = 250$  and  $L_o = 500$  m, Fig.6a), but the overall pattern is similar. I added references to Yoon et al. and Kohout et al. to the revised manuscript.

40. page 10, line 12: This statement seems to hold when the floe length is less than half the wavelength. Maybe the author could mention that.

As Fig.6b shows, the response of floes smaller than half the wavelength is almost exactly symmetrical, with maximum stress in the middle of the floe. When the floe length increases beyond half the wavelength, the symmetry gradually vanishes, but at first no second maximum is present – see the curve for  $L_o = 25$  m (the wavelength in this figure is 42 m). I added this information to the modified text.

41. page 10, line 17: I don't understand the statement "An individual wave is responsible for a few breaking events". Could the author clarify?

Take a closer look at Figs. 7b and 9b; for example, in 7b, two or three breaking events (pink dots) take place within one wave period (graphically, within one inclined yellow stripe representing a wave propagating into the ice).

42. page 10, line 18: what is meant by "weaker ice"? Is it small thickness, tensile strength, ...?

I wrote it explicitly: "In thinner ice..."

43. page 11, line 1: can the author define or at least briefly describe a "Jonswap energy spectrum"?

I added the text "(one of widely used idealized models of wave energy spectra, suitable for a wide range of wind and fetch conditions)" to this sentence.

44. page 11, line 15: I do not agree that breaking does not depend on "the characteristics of the incoming waves", as clearly it will depend on wave amplitude. We can TCD
probably expect a linear relationship between incident wave amplitude and stress. This comment also applies to a similar statement in line 27.

The amplitude is important only in the sense that it must exceed certain threshold. Breaking occurs as soon as the stress exceeds the strength of the ice – it cannot reach higher values. See also my comments below (No. 45 and 50).

45. page 11, lines 17 and 18: main result (iii) is rather obvious as there is not much else the stress can depend on!

I agree that taken alone it is quite obvious – but it should be read together with result (ii): the wave characteristics decide *if* the ice break, but not *where* it breaks.

46. page 11, lines 19 and 20: the author has not introduced the concept of "breaking probability" earlier in the manuscript, so it is confusing to have it in the conclusion. Maybe this could be slightly rephrased.

Ice breaking probability is mentioned in the introduction in relation to the work of Vaughan and Squire (2011), but you are right, this term has not been properly defined. I think it is clear in the context of the paragraph in which it is used, but to avoid confusion I changed it to a simple statement: "... and thus breaking is most likely to occur."

47. page 11, line 22: replace "might be not realistic" by "might not be realistic".

I wrote "might be unrealistic" instead.

48. page 11, lines 27 and 28: I think the author should also mention multiple scattering here, as discussed in my earlier comment.

This part of the discussion has bene largely modifed/rewritten – see the revized manuscript.

49. page 11, lines 27-29: can the author include a reference to support the statement in brackets?

TCD
It is hard to include just one (or even a few) references, as the great majority of studies of FSD in MIZ demonstrate the existence of wide, heavy-tailed FSDs. I added reference to Toyota et el. 2011 and 2016, as they are recent and themselves contain a lot of references to papers on FSD.

50. page 11, lines 29-32: I am having trouble agreeing with this statement, as I think the incident wave amplitude plays a determining role in creating the floe size distribution, i.e. the larger the wave amplitude the smaller the floes.

The larger the wave amplitude the smaller the floes... If one consideres a whole range of breaking mechanisms that contribute to breaking and therefore influence the FSD (floe–floe collisions, overwash, rafting, etc.) then certainly: yes. But one should keep in mind how the parameterizations of wave-induced breaking for large-scale sea ice models work: it is assumed that the FSD has a prescribed shape (e.g., a power law with a certain slope) and *only the maximum floe size*  $L_{max}$  *is allowed to vary*.  $L_{max}$  is related to bending stresses (all floes larger than  $L_{max}$  break due to bending). All other breaking mechanisms that I mentioned above and that – as you suggest – are wave amplitude dependent, are included in the power-law part of the FSD, as it is assumed that they produce scale-invariant pdfs of floe sizes.

The results of the present model (and the earlier ones by Squire and colleagues) suggest that  $L_{max}$  should be a function of ice properties, not wavelength. If bending stresses are concerned, the only relevant thing is whether the wave amplitude exceeds a threshold required for breaking or not. This is what I had in mind when I wrote this sentence ("the information on incoming waves is used to determine whether breaking of ice takes place"). But I agree that the wave amplitude (or rather: steepness) has influence on FSD as a whole, and presumably modifies the exponent of the distribution.

In the revised manuscript, I modified/extended the statement in question in order to make it more clear.
**REPLY TO THE COMMENTS OF REVIEWER No. 2**

2.1. p2: 'For example, the functional form describing the rate of change of wave height with distance from the ice edge is far from established...' Meylan et al (2014) found exponential decay from this dataset when the individual frequencies are considered instead of the significant wave height. Li et al (2015) were able to qualitatively produce the linear decay of the significant wave height with Wavewatch 3 – proposing that the nonlinear source term  $S_{nl}$  was the reason, as it moves energy to lower frequencies which are attenuated more slowly.

As I wrote in the general comments above, I decided to remove this text from the revised manuscript.

2.2. p8: please give the equations for the free-slip boundary conditions.

Yes, thank you for pointing this out. The free-slip boundary conditions at the bottom were given in Eq. (6), and they should be written together with Eqs. (21)-(22) and (23)-(24) as well (for the horizontal and vertical velocity components, respectively). I have added the missing formulae, and rearranged the order of equations so that they directly correspond to (5)-(7) in Section 2.2.1.

2.3. equation (7): Ma et al (2012) has the total derivative of w in the bottom condition (dw/dt). Maybe this is a typo? Similarly, perhaps the pressure condition at the lower surface of the ice (21) should be checked?

Yes, the first mistake is a typo, the second one – an effect of copying/pasting this formula... Thank you for noticing it!

2.4. p6: How are the waves generated?

The last sentence in section 2.2.1 says: "In the model applications presented in this work, sponge layers are applied at the left and right boundary, and waves are generated inside the model domain (Ma et al., 2014)." Both things – sponge layers and internal wave generation – are working exactly as described in the NHWAVE technical
documentation and in papers describing the model. These parts of the NHWAVE code are unaffected by the coupling with the ice module. In the revised manuscript, I added a short comment to the above sentence to make it more clear.

**2.5. §3.1 Stress variability in continuous ice**

(i) The locations of  $\sigma_{t,max}$  seem to occur about 10m away from the ice edge, which seems very small, eg. Squire et al (1995): "Anecdotally it appears that incoming waves and swell cause a fracture line to develop a few tens of meters back from the ice margin and parallel to it". Was there any attempt to tune the wave number in the ice to a realistic value, eg. for a thin elastic plate? Perhaps these numbers would increase if such tuning were done.

(ii) Presumably the damping of the waves is due to some damping in the bonds, although this is not mentioned anywhere. Perhaps if this was removed it could be useful to see how much of the attenuation is numerical and how much is physical. The reflection and transmission coefficients could also be compared to linear models for a semi-infinite elastic plate. It would also be interesting to see if this affects the conclusions about the location of  $\sigma_{t,max}$  being independent of wavelength or not.

See my general comments on the damping of wave energy in the model and on the choice of the model parameters.

**2.6. §3.2 Breaking of uniform ice by regular waves**

(i) It would be interesting to see fig 8 without damping in the bonds and also the time evolution of the wave amplitude to see if there is more or less attenuation after breaking – on one hand the floes produced are small compared to the wavelength but on the other hand there are many of them and perhaps multiple scattering could do something.

Again: see my previous comments. Also, I think that Fig. 9 (previous Fig. 8) at least partly answers your question, as it compares the wave amplitude in simulations without and with breaking.
**REPLY TO THE COMMENTS OF REVIEWER No. 3 (A. Kulchitsky**

2.1.1. My main concern about this work is the lack of verification of the approach and the code. For example, does discretization step  $\Delta x$  change the outcome of the computations? It is unclear until some scaling tests are done. I would suggest to use different  $\Delta x$  with the same wave input and compare the results. This can be corrected within this paper by running some tests with  $\Delta x$  being a half of the  $\Delta x$  used in the paper and comparing the results. Maybe a reference that the final results do not change with  $\Delta x$  within the numerical accuracy would be sufficient.

I fully agree that some comments on the influence of the model resolution on the results should be added to the manuscript. For the simulations presented in this work, the ratio of  $\Delta x$  to the length of the waves is crucial and should not exceed a certain limiting value (this is true for standalone NHWAVE model as well). Similarly, the number of  $\sigma$ -layers,  $N_l$ , must be sufficient to capture the vertical structure of the waves. Before running the model for the cases described in the manuscript, several "verification runs" were conducted. In computations without ice, the wave model with prescribed  $\Delta x$  and  $N_l$  was run for a set of wavelengths in order to estimate the shortest wave that can propagate in the model without artificial damping related to insufficient model resolution (we expect zero attenuation, because the dissipation was turned off). The shortest waves considered in the "proper" simulatons are longer than the limiting wavelength found in this way.

As for the coupled waves+ice simulations, the situation is more complex. In all DEM models, the macroscopic properties of the modelled material are sensitive not only to microscopic properties of bonds connecting the grains, but also to the grain size distribution. This is well known, and selecting grain size is a standard part of the process of calibration of a DEM model to a particular situation – see, e.g., Potyondy and Cundall, Int. J. Rock Mech. Mining Sci., 2004; Koyama and Jing, Engng Analysis with Bound. Elem., 2007; there many other papers on the subject. Generally, changing the size of grains changes the macroscopic behavior of the modelled material, or, in

TCD
other words, the grain size is not a free parameter that can be varied without affecting the properties of the material. In our case, if  $\Delta x$  is changed, the ice response to waves changes. Or, *vice versa*, in order to have the same ice response with different  $\Delta x$ , the model parameters have to be adjusted. In the study on which I am working on right now, the model is going to be applied to a laboratory setting, in which a continuous ice sheet was broken by regular waves. Calibration and validation will be a crucial element of the modelling process. In the work presented in this manuscript, however, no attempt to reproduce any particular real-world situation was made, and therefore the role of  $\Delta x$  is not analyzed.

I added information on that to the first part of section 3 of the revised manuscript.

2.1.2. I have some concerns regarding 2D approach to the problem of breaking an ice floe. It is just hard to imagine how to interpret the results of a single ice flow breaking in long (y-dimension) but narrow (x-dimension) fragments as the results of simulations show. Actual floe would break along x dimension as well such that it will be easier to break later. I suppose 2D approach is still applicable for testing in channels. This issue is not to be corrected and just brought to the discussion at this point.

Yes, the fact that the model has only one horizontal dimension is an important limitation. Even though many observations show that the shape of ice floes detached from the edge of a continuous ice sheet by swell waves can be very elongated, i.e., the floes' size in the wave propagation direction is much smaller than their size in the perpendicular direction, obviously sea ice breaking by waves takes place in two horizontal dimensions. This fact is related to nonuniformities of the sea ice properties as well as to the directionality of the incomming waves (both the mean wave direction, which doesn't have to be perpendicular to the ice edge, and directional spreading). These effects cannot be taken into account in the present model, but the 2D-V formulation described in this manuscript should be treated as a "proof of concept" for a more general model based on NHWAVE and DESIgn. TCD
2.2.1. I struggled to understand the 2.1 and 2.2 sections with just Figure 1. It would be very useful to add dimensions, indices and more notes to the figure. In this case the reader would see what  $\Delta x$ ,  $l_i$ , etc. are. Maybe a single cell scheme with forces and torques shown, as well as velocities, constraints ( $u_i = 0$ ) and bonds.

As already mentioned in my general comments, I added a new figure showing a sketch of two grains connected with a bond, with relevant dimensions of these objects etc. I hope this makes the description of the model components in section 2 easier to understand. I also added the axes (x,z) to Fig.1.

2.2.2. Sec. 2.1, p. 5, 0-5. "All bonds are cuboid". Should it say "All bonds are elastic"?

No. This statement refers to the shape of the bonds, not their material properties.

2.2.3. Equations (13) and (14) include terms for the torques and forces imposed from water ( $M_{wv,i}$  and  $F_{wv,i}$ ) that are only explained 2 pages later in the coupling section. They should be defined after the equations as well.

These terms are explained in Section 2.2.2: "Finally, the first terms on the right-handside of (13) and (14) denote the net moment of forces and the net vertical force, respectively, from the wave motion underneath the ice. They are calculated by integrating the contribution from waves over the wetted surface of the grains. Their detailed formulation is given further in Section 2.2.3." In the revised manuscript, I moved this sentence to the paragraph that directly follows Eqs. (11)–(17). I agree that it should have been there from the start.

2.2.4. I have some questions regarding "classical beam theory" equations (18)-(20). First, the length of the bonds is defined generally as  $l_{b,i} = \lambda \Delta x$  (page 5, 0-5). Please, verify that  $l_{b,i} = h_{b,i}$  for these computations. Moreover, I suggest to introduce a picture showing all the stresses on the bonds.

No,  $l_{b,i}$  does not equal  $h_{b,i}$ . The first is the bond dimension in the *x*-direction, the second – in the *z*-direction. Thus,  $h_{b,i}$  corresponds to ice thickness and, as the text

TCD
in the first part of Section 3 says, in the simulations presented in the manuscript the thickness of grains and bonds was equal.

In equations (18)-(20), the forces are divided by the bond cross-sectional area, but as the width of the bonds (and grains, for that matter) equals 1, only  $h_{b,i}$  is left.

As for the stresses acting on the bonds, the new Fig. 2 shows the locations of the maximum compressive and tensile stresses acting on the bond and resulting from relative rotation marked in red.

---

## Referee Report (RR1)

Review of the revised manuscript *Wave-induced stress and breaking of sea ice in a coupled hydrodynamic–discrete-element wave–ice model* by A. Herman

My comments and suggestions were addressed appropriately by the author and I support publication of the manuscript. I respect the author's decision not to follow some of my suggestions, as it is her paper and should decide what general form the investigation takes. I think this is a significant piece of work which may attract much attention in the field of wave/ice interactions, and as such it is important to make it as easy as possible for the reader to go through the paper. My suggestions, e.g. comments 3 and 4 of the original review, were only made to help improve the clarity and readability of the paper in this respect. A few of my original points also seem to have been misunderstood by the author, so I clarify them below and suggest that the author makes further minor revisions in response. A few other minor comments are also detailed below.

**Comments**

1. What I meant regarding the limitation of "small water depth" is that the all the simulations are done for a water depth of $10\,\mathrm{m}$ which is relatively small for the range of wave periods considered. In particular, the effect of water depth on the wavelength in open water is important for wave periods larger than approximately $5\,\mathrm{s}$. I understand that no shallow-water approximation was used here. I simply wonder whether if it would be computationally feasible to perform simulations with larger water depth of $\mathrm{O}(100\,\mathrm{m})$, for which deep water conditions are well approximated. The author partially addressed this in the response to my comment, but I think this should also be discussed in the manuscript, probably at the end of section 4.

2. The first two paragraphs of the Introduction are much improved. My original comment regarding the second paragraph mainly concerned the beginning of the paragraph and especially the long list of references of previous work on many aspects of wave/ice interactions. It is disconnected from the list of topics studied in this area provided earlier in the paragraph. In particular, it is unclear which paper studies what aspect of the problem. I suggest merging the two lists, i.e. topics and references, so an appropriate reference is given for each topic when it is mentioned. Also, there is an inconsistency in the reference Montiel et al. (2016). The corresponding item in the reference list is a paper published in *Annals of Glaciology* in 2015, not 2016. I assume the author intended to, or otherwise should, reference the paper by the same authors published in *Journal of Fluid Mechanics* in 2016, which is a generalisation of that of 2015.

3. Page 2, line 3: a more recent release of WW3 (The WAVEWATCH III Development Group, 2016) with more ice parametrisations is now available and should therefore replace

the reference to that of 2014.

4. Page 2, lines 7, 8: "Due to low temporal resolution of satellite data in polar regions, they provide only snapshots of sea ice conditions" needs to be rephrased as it does not read well. I suggest: "Due to their low temporal resolution in polar regions, satellite data only provide snapshots of sea ice conditions, ...". A reference for recent advances in monitoring waves in the MIZ should be added, e.g. Ardhuin et al. (2017, *Remote Sensing of Environment*).

5. Page 2, line 12: is missing from "one of them IS ice breaking".

6. Page 2, line 24: it will be unclear to most readers what "secondary ice-coupled waves" means. I suggest removing this or explaining more. I think the "basic mechanisms of wave-induced ice breaking" can be more simply described as flexural failure, as it is not mentioned before in the text that ice floes do flex under wave action. An explanation of the "secondary ice-coupled wave" can be given in page 13, line 30. The key here is that this damped oscillatory mode constructively interferes with the travelling mode (or primary wave) to give the maximum in strain at some small distance from ice edge.

7. Page 5, line 5: the observation that surge and drift are negligible in compact sea ice is not likely to be known and understood by all readers, so a reference may be useful here or additional explanation. I imagine the reason why a compact broken ice cover does not show surge motion is that individual floes are in contact and therefore continually collide, therefore restricting their ability to surge.

8. Page 10, line 12: replace "they" by "that" or "which".

9. Page 12, line 4: I think "an individual wave" is slightly confusing. I suggest that the sentence is rephrased to clarify that the several breakup events occur within one wavelength.

10. Page 13, section 4: I'm having trouble getting my head around the statements that the location of breaking is independent from wave characteristics and that amplitude only acts as a switch that decides whether or not breaking takes place. As I understand it, the author's argument is that breakup occurs where the primary and secondary ice-coupled waves constructively interact to produce a maximum in strain at a small distance from the ice edge. Changing the wave amplitude does not change the location of this maximum; I agree with that. However increasing the amplitude beyond a certain threshold will result in a situation where the primary travelling wave causes breakup well beyond this local strain maximum in a region where the secondary wave is insignificant. The primary wave will only attenuate due to dissipative processes or multiple scattering, which are slow attenuation mechanisms. Wave breakup events were reported hundreds of kilometres from the ice during the SIPEX-2 voyage in the Ross Sea as a result of large amplitude storm waves travelling through the MIZ almost unattenuated (see Kohout et al., 2016, *Deep Sea Research II*). This breaking mechanism is not captured by the present model and I think the author should simply acknowledge this in their conclusion.

11. Page 14, line 15: "directional with" should be "directional width" I imagine.

12. Page 14, line 23: replace "are" by "is".

---

## Referee Report (RR2)

Thank you for carefully addressing my comments and improving the text.

However, I disagree with the statements about the necessary dependence of the overall process on $\Delta x$. Moreover, I think this statement is actually makes the DEM approach questionable unless fixed.

The author says in Section 3.1 (30): "It is also worth stressing that — as in all DEM models — the macroscopic properties of the modeled material (its strength, elastic modulus, and so on) depend not only on the microscopic properties of grains and bonds, but also on the grain size (e.g., Potyondy and Cundall, 2004; Koyama and Jing, 2007)."

I do not think this statement is true. Potyondy and Cundall say, for example, that the local properties like cement modulus are dependent on grain size in their model to actually achieve the macro properties that are independent on size. Sure, this means that the local coefficients must be selected properly to achieve the overall convergence when the grain size changing but it shows their effort to keep the macro properties controlled. I actually think even this approach is not well justified as physical laws should be formulated such that coefficients are size independent.

For example, [Kumar et al., 2016] prefer not to use normal or tangential stiffnesses as parameters for local models but talk about material (contact) parameters as they introduce:

$$\Sigma_N = K_N/2R^*, \quad \Sigma_T = K_T/2R^*,$$

in their laws making the contact laws independent on block size. Where the normal and tangential stiffnesses $K_N$ and $K_T$ are size dependent as necessary, and $R^*$ is the effective radius at the contact point.

Different authors use different approaches to ensure that the global parameters do not change with discretization which is essentially are ice blocks in author's model. For example, [Leclerc, 2017] carefully calibrate the beam model in their DEM approach to match macro properties by running their model for the large range of local parameters. They ensure that the DEM model accurately reproduce elastic behavior of the material comparing the results with finite elements analysis.

I am familiar with the same approach to match adhesion local properties such as surface energy $\gamma$ to achieve the correct bulk material adhesive strength. For example, [Kulchitsky et al., 2016] also use both theoretical consideration and extensive calibration on known engineering tests to connect local contact properties with macro parameters before they use DEM to do the quantitative comparisons.

The grain size distribution is also important in some processes as you say but it does not mean that the mean value has to be exactly the same as in the actual material to achieve the right physical results. I prefer to say that *the grain size is a resolution of the model and hence the macro physics must not depend on it* unless you actually work with the grains that exactly match the grains in your physical process. If the resolution becomes "a free model parameter", there is something wrong with such a model.

Overall, I think it is very fruitful to think about DEM with bonds models solving actual classical elastic or other continuous problems until finite deformation occur. As it can be seen in [Leclerc, 2017] or [Jin et al., 2011], DEM actually reproduce elastic behavior before the bonds are broken if the macro parameters are well calibrated with the local contact properties or even better local laws are correctly chosen.

For the ice model the ice blocks have very regular cuboid shapes. In this case the local contact laws should contain the dimension explicitly and can be formulated using coefficients that are

independent on the block size. For example, the stiffness coefficient can be related to the contact area between the blocks somehow.

   *Minor 2.2.4 reply:* That is the only thing I asked if $l_{b,i} = h_{b,i}$ in the actual computations. Apparently, that is the case. Actually, I noticed this because often it is very useful to make test computations with two bond dimensions being not equal to each other to see if there are any problems in the algorithm implementation or dependence on their ratio.

**References**

[Jin et al., 2011] Jin, F., Zhang, C., Hu, W., and Wang, J. (2011). 3d mode discrete element method: Elastic model. *International Journal of Rock Mechanics and Mining Sciences*, 48(1):59–66.

[Kulchitsky et al., 2016] Kulchitsky, A. V., Johnson, J. B., and Reeves, D. M. (2016). Resistance forces during boulder extraction from an asteroid. *Acta Astronautica*, 127:424–437.

[Kumar et al., 2016] Kumar, R., Rommel, S., Jauffrès, D., Lhuissier, P., and Martin, C. L. (2016). Effect of packing characteristics on the discrete element simulation of elasticity and buckling. *International Journal of Mechanical Sciences*, 110:14–21.

[Leclerc, 2017] Leclerc, W. (2017). Discrete element method to simulate the elastic behavior of 3d heterogeneous continuous media. *International Journal of Solids and Structures*.

---

## Author Response (AR2)

Response to reviewes
and a marked-up version of the manuscript
"Wave-induced stress and breaking of sea ice
in a coupled hydrodynamic–discrete-element
wave–ice model"

Agnieszka Herman

September 28, 2017

**RESPONSE TO THE COMMENTS OF FABIEN MONTIEL**

1. I added the text: "Although in the computations presented in this paper the water depth was relatively shallow ($H$=10 m), deep-water waves can be simulated without significant increase in computational costs, because the model enables non-equally spaced σ-layers, with thickness adjusted to the vertical structure of the wave." to the last paragraph of the discussion section (see the revised manuscript).

2. I understand the criticism, but as many of the papers I cite in paragraph 2 are related to more than one topic, it is not easy to assign them unambiguously to just one topic on the list. I tried to do this, but as this would require citing some of the papers more than once, the result was not clearer than the present version of the text.
As far the citation to Montiel et al. (2016) is concerned: yes, I meant the JFM paper from 2016, the mistake resulted from mismatched labels in my BibTeX file (and incorrect publication year of the AG paper). I corrected this. Thank you for pointing this out!

3. I updated the reference as suggested.

4. I corrected the sentence and added the suggested reference.

5. Yes, I corrected this.

6. I provide references in which the readers can find definition/explanation of ice-coupled waves, so I don't think that using this term is problematic.

7. I added the text: "(obviously, this is true in a continuous, unbroken ice sheet; in broken ice at high ice concentration, i.e., with densely packed floes, horizontal motion is suppressed by collisions between neighbouring floes)" to the sentence in question.

8. Corrected.

9. I think this expression is more accurate than the suggested one. The wave is traveling into the ice and the breaking events do not necessarily occur "within one wavelength" from the ice edge – especially that, as this paper shows, the wavelength does not determine where breaking occurs.

10. But to my understanding, even in the situation described in the last part of your argument, breaking would occur progressively from the ice edge towards farther and farther regions of the ice sheet – and not rapidly everywhere. How can a high-amplitude wave propagate far into the ice without first breaking it closer to the ice edge? Unless some mechanisms are there leading e.g. to the increasing steepness of the waves within the ice; or the ice is spatially nonuniform and breaks in certain areas more easily than in others. But still, once the first cracks form, I believe the mechanism described in this paper will contribute to propagation of fracture further from that zone.

11. Of course. I corrected this.

12. I think this is correct: the computational costs ARE a limitation. Why "is"?

**RESPONSE TO THE COMMENTS OF ANTON KULCHITSKY**

Thank you for very detailed comments and an overview of different approaches to the treatment of micro/macro scale relationships in DEM models. I think that most of your concerns result from misunderstanding of my statements in the paper – which is my fault, of course, as I should have been more careful in phrasing of those statements.
I definitely did not mean that one doesn't have control over macroscopic properties of the material modelled with DEM. The macroscopic properties can be controlled and kept constant with changes of the grain size, but – as you write in the context of the work of Potyondy and Cundall – "the local coefficients must be selected properly to achieve the overall convergence when the grain size changing". This is exactly what I meant. I fully agree that "careful calibration" of the model parameters is necessary if the model is expected to reproduce the macroscopic behavior of the analyzed material. And that the degree to which the macroscopic properties depend on the microscopic ones depends on the details of the contact and bond models used.

I reformulated the controversial paragraph – it has become shorter, but I hope its message is now clear.

**RESPONSE TO THE COMMENTS OF THE ANONYMOUS REVIEWER**

(1) Corrected (in the suggested location and in the abstract, where the same expression is used).
(2) The second sentence was added following the suggestion of another reviewer (Fabien Montiel). As it is indeed controversial, I decided – after discussing the matter with colleagues and after reading your comment - to remove it and to return to the first version of the text.
(3) I added the comment: "see also Fig. 4 and Section 3.1 for the discussion on the sources of damping in the present model version" to the text describing Fig. 7 (see the revised manuscript).

I corrected all typos as suggested. The units of $z_i$ and $\sigma_t$ have been added to the figure captions.

[revised manuscript text omitted]